# A non-conducting role of the Ca$_v$1.4 Ca$^{2+}$ channel drives homeostatic plasticity at the cone photoreceptor synapse

J Wesley Maddox[1†], Gregory J Ordemann[1†], Juan AM de la Rosa Vázquez[1], Angie Huang[1], Christof Gault[1], Serena R Wisner[2,3], Kate Randall[1], Daiki Futagi[4], Nihal A Salem[1], Dayne Mayfield[1], Boris V Zemelman[1], Steven DeVries[4], Mrinalini Hoon[2,5], Amy Lee[1*]

[1]Department of Neuroscience, University of Texas-Austin, Austin, United States; [2]Department of Ophthalmology and Visual Sciences, University of Wisconsin-Madison, Madison, United States; [3]Neuroscience Training Program, University of Wisconsin-Madison, Madison, United States; [4]Department of Ophthalmology, Northwestern University Feinberg School of Medicine, Chicago, United States; [5]McPherson Eye Research Institute, Madison, United States

*For correspondence:
amy.lee1@austin.utexas.edu

†These authors contributed equally to this work

Competing interest: The authors declare that no competing interests exist.

## eLife assessment

Based on analyses of retinae from genetically modified mice, and from wild-type ground squirrel and macaque, employing microscopic imaging, electrophysiology, and pharmacological manipulations, this **valuable** study on the role of Cav1.4 calcium channels in cone photoreceptor cells (i) shows that the expression of a Cav1.4 variant lacking calcium conductivity supports the development of cone synapses beyond what is observed in the complete absence of Cav1.4, and (ii) indicates that the cone pathway can partially operate even without calcium flux through Cav1.4 channels, thus preserving behavioral responses under bright light. The evidence for the function of Cav1.4 protein in synapse development is **convincing** and in agreement with a closely related earlier study by the same authors on rod photoreceptors. The mechanism of compensation of Cav1.4 loss by Cav3 remains unclear but appears to involve post-transcriptional processes. As congenital Cav1.4 dysfunction can cause stationary night blindness, this work relates to a wide range of neuroscience topics, from synapse biology to neuro-ophthalmology.

**Abstract** In congenital stationary night blindness, type 2 (CSNB2)—a disorder involving the Ca$_v$1.4 (L-type) Ca$^{2+}$ channel—visual impairment is mild considering that Ca$_v$1.4 mediates synaptic release from rod and cone photoreceptors. Here, we addressed this conundrum using a Ca$_v$1.4 knockout (KO) mouse and a knock-in (G369i KI) mouse expressing a non-conducting Ca$_v$1.4. Surprisingly, Ca$_v$3 (T-type) Ca$^{2+}$ currents were detected in cones of G369i KI mice and Ca$_v$1.4 KO mice but not in cones of wild-type mouse, ground squirrels, and macaque retina. Whereas Ca$_v$1.4 KO mice are blind, G369i KI mice exhibit normal photopic (i.e. cone-mediated) visual behavior. Cone synapses, which fail to form in Ca$_v$1.4 KO mice, are present, albeit enlarged, and with some errors in postsynaptic wiring in G369i KI mice. While Ca$_v$1.4 KO mice lack evidence of cone synaptic responses, electrophysiological recordings in G369i KI mice revealed nominal transmission from cones to horizontal cells and bipolar cells. In CSNB2, we propose that Ca$_v$3 channels maintain cone synaptic output provided that the nonconducting role of Ca$_v$1.4 in cone synaptogenesis remains intact. Our findings reveal an unexpected form of homeostatic plasticity that relies on a non-canonical role of an ion channel.

## Introduction

At the first synapse in the visual pathway, graded electrical signals produced in rod and cone photoreceptors gate the release of glutamate onto postsynaptic neurons. Photoreceptor synapses are specialized with a vesicle-associated ribbon organelle and postsynaptic neurites of horizontal and bipolar cells that invaginate deep within the terminal (*Haverkamp et al., 2000*). A variety of proteins interact with the ribbon and synaptic vesicles near release sites (i.e. active zones) (*Mercer and Thoreson, 2011*). The importance of these proteins for vision is illustrated by the numerous inherited retinal diseases linked to mutations in their encoding genes (*Frederick and Zenisek, 2023*).

One such gene is *CACNA1F*, which encodes the voltage-gated $Ca^{2+}$ ($Ca_v$) channel expressed in retinal photoreceptors, $Ca_v1.4$ (*Mansergh et al., 2005*; *Liu et al., 2013*; *Chang et al., 2006*). Among the sub-family of $Ca_v1$ L-type channels, $Ca_v1.4$ exhibits unusually slow inactivation that is well-matched for supporting the tonic, $Ca^{2+}$-dependent release of glutamate from photoreceptor synaptic terminals in darkness (*Singh et al., 2006*; *Wahl-Schott et al., 2006*). More than 200 mutations in *CACNA1F* cause vision disorders including congenital stationary night blindness type 2 (CSNB2) (*Bech-Hansen et al., 1998*; *Strom et al., 1998*). These mutations are broadly categorized as producing a gain of function or loss of function in $Ca_v1.4$ (*Waldner et al., 2018*). How these mutations in *CACNA1F* lead to the variable clinical phenotypes of CSNB2 is largely unknown. Symptoms may include strabismus, low visual acuity, and in many cases, night blindness (*Boycott et al., 2000*; *Zeitz et al., 2015*). The latter suggests a primary defect in rod pathways, which is surprising given that $Ca_v1.4$ KO mice are completely blind and lack any evidence of either rod or cone synaptic responses (*Mansergh et al., 2005*; *Liu et al., 2013*; *Regus-Leidig et al., 2014*). A major caveat is that rod and cone synapses do not form in $Ca_v1.4$ KO mice (*Liu et al., 2013*; *Regus-Leidig et al., 2014*; *Zabouri and Haverkamp, 2013*; *Raven et al., 2008*). Thus, $Ca_v1.4$ KO mice are not suitable for studies of how *CACNA1F* mutations differentially affect rod and cone pathways or for efforts to uncover how the biophysical properties of $Ca_v1.4$ shape photoreceptor synaptic release properties.

Here, we overcome this hurdle with a knock-in mouse strain (G369i KI) expressing a non-conducting mutant form of $Ca_v1.4$ (*Maddox et al., 2020*). We show that, although greatly impaired, cone synapses and downstream signaling through cone pathways can support photopic visual function in G369i KI mice. This novel mechanism requires the ability of the $Ca_v1.4$ protein, independent of its $Ca^{2+}$ conductance, to nucleate the assembly of cone ribbon synapses and involves an aberrant $Ca_v3$ (T-type) conductance that appears when $Ca_v1.4$ $Ca^{2+}$ signals are compromised.

## Results

### Cone pedicles and ribbons are present in G369i KI mice but not in $Ca_v1.4$ KO mice

The G369i mutation results in an insertion of a glycine residue in a transmembrane domain, which prevents $Ca^{2+}$ permeation through $Ca_v1.4$ (*Maddox et al., 2020*). The presence of rod synapses in G369i KI mice but not $Ca_v1.4$ KO mice indicates that the $Ca_v1.4$ protein and not its $Ca^{2+}$ conductance is required for rod synapse assembly (*Maddox et al., 2020*). To test if this is also true for cones, we performed immunofluorescence and confocal analyses using antibodies against cone arrestin (CAR) and CtBP2 to label cone terminals (i.e. pedicles) and ribbons, respectively (*Figure 1*). In $Ca_v1.4$ KO mouse retinas, cone pedicles were shrunken and retracted into the outer nuclear layer (ONL, *Figure 1a*) with little evidence of elongated ribbons (*Figure 1b*). In contrast, cone pedicles in G369i KI mice were normally localized in the outer plexiform layer (OPL, *Figure 1a*) and were populated by multiple ribbons that were associated with labeling for $Ca_v1.4$ (*Figure 1b*). Some abnormalities were apparent in G369i KI cone pedicles, such as extremely elongated ribbons and telodendria extending toward the inner nuclear layer (*Figure 1b*). These results indicate that $Ca_v1.4$ $Ca^{2+}$ signals are dispensable for the integrity of ribbons and the pedicle but are necessary for their overall refinement.

### $Ca_v3$ channel activity is present in cones of G369i KI and $Ca_v1.4$ KO mice but not WT mice, ground squirrels, or macaque retina

A prevailing yet unsupported hypothesis regarding the relatively mild visual phenotypes in CSNB2 is that additional $Ca_v$ subtypes may compensate for $Ca_v1.4$ loss-of-function in cones. If so, then $Ca^{2+}$

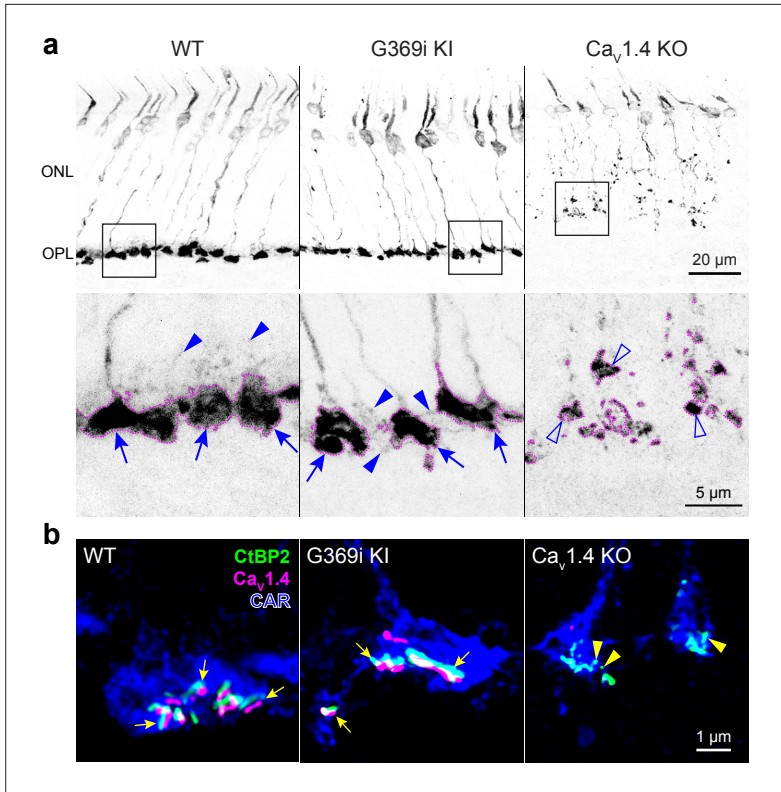

**Figure 1.** Cone pedicles and the mutant Ca$_v$1.4 G369i channel are normally localized in the retina of G369i KI mice but not Ca$_v$1.4 KO mice. Confocal images of the outer nuclear layer (ONL) and outer plexiform layer (OPL) of retina from wild-type (WT), G369i KI, and Ca$_v$1.4 KO mice that were labeled with antibodies against cone arrestin (CAR), CtBP2, and Ca$_v$1.4. (**a**) Inverted images of CAR labeling. Lower panels depict pedicles labeled by CAR antibodies (dotted outlines) and correspond to the boxed regions in the upper panels. Cone pedicles remain in the OPL of WT and G369i KI retina (arrows) but are misshapen and retracted into the ONL of the Ca$_v$1.4 KO retina (open arrowheads). Solid arrowheads indicate telodendria which extend only apically in the WT retina but extend basally and laterally in the G369i KI retina. (**b**) Deconvolved confocal images showing Ca$_v$1.4 labeling near cone ribbons in WT and G369i KI pedicles (arrows) and ribbon spheres without Ca$_v$1.4 labeling in the Ca$_v$1.4 KO pedicle (arrowheads). Rod-associated CtBP2 and Ca$_v$1.4 labeling was removed for clarity.

currents ($I_{Ca}$) mediated by these subtypes should be evident in cones of Ca$_v$1.4 KO and G369i KI mice. To test this, we performed patch clamp recordings of cones in retinal slices of adult WT, G369i KI, and Ca$_v$1.4 KO mice under conditions designed to isolate $I_{Ca}$. During voltage ramps in WT cones, $I_{Ca}$ activated around –50 mV and peaked near –20 mV, consistent with the properties of Ca$_v$1.4 (**Figure 2a–d**). While not observed in rods of G369i KI or Ca$_v$1.4 KO mice (**Maddox et al., 2020**), a small-amplitude $I_{Ca}$ that activated around –60 mV and peaked near –35 mV was detected in cones of these mice (**Figure 2a–d**). To uncover this low voltage-activated $I_{Ca}$, we used a faster voltage ramp (0.5 mV/ms) than in the previous study of G369i KI rods (~0.15 mV/ms) **Maddox et al., 2020**. With this faster voltage ramp, $I_{Ca}$ was not apparent in G369i KI rods (**Figure 2—figure supplement 1**). Therefore, the aberrant $I_{Ca}$ is an adaptation to Ca$_v$1.4 loss-of-function in cones but not rods.

We next used step depolarizations to identify the type of Ca$_v$ channel underlying $I_{Ca}$ in G369i KI and Ca$_v$1.4 KO cones. Analysis of current-voltage (I-V) and conductance-voltage (G-V) relationships revealed a significant, hyperpolarizing shift (~10 mV) in the voltage of half-maximal activation ($V_h$) in G369i KI cones (**Figure 2c and d**; **Table 1**). $I_{Ca}$ in G369i KI cones was not sustained as in WT cones but inactivated rapidly during 500 ms step depolarizations (**Figure 2e**). For G369i KI cones, the overlay of the conductance-voltage (**Figure 2d**) and inactivation curves revealed a sizeable window current (**Figure 2f**). These features of $I_{Ca}$ in G369i KI and Ca$_v$1.4 KO cones resembled those of Ca$_v$3 T-type channels rather than Ca$_v$1.4 **Perez-Reyes, 2003**.

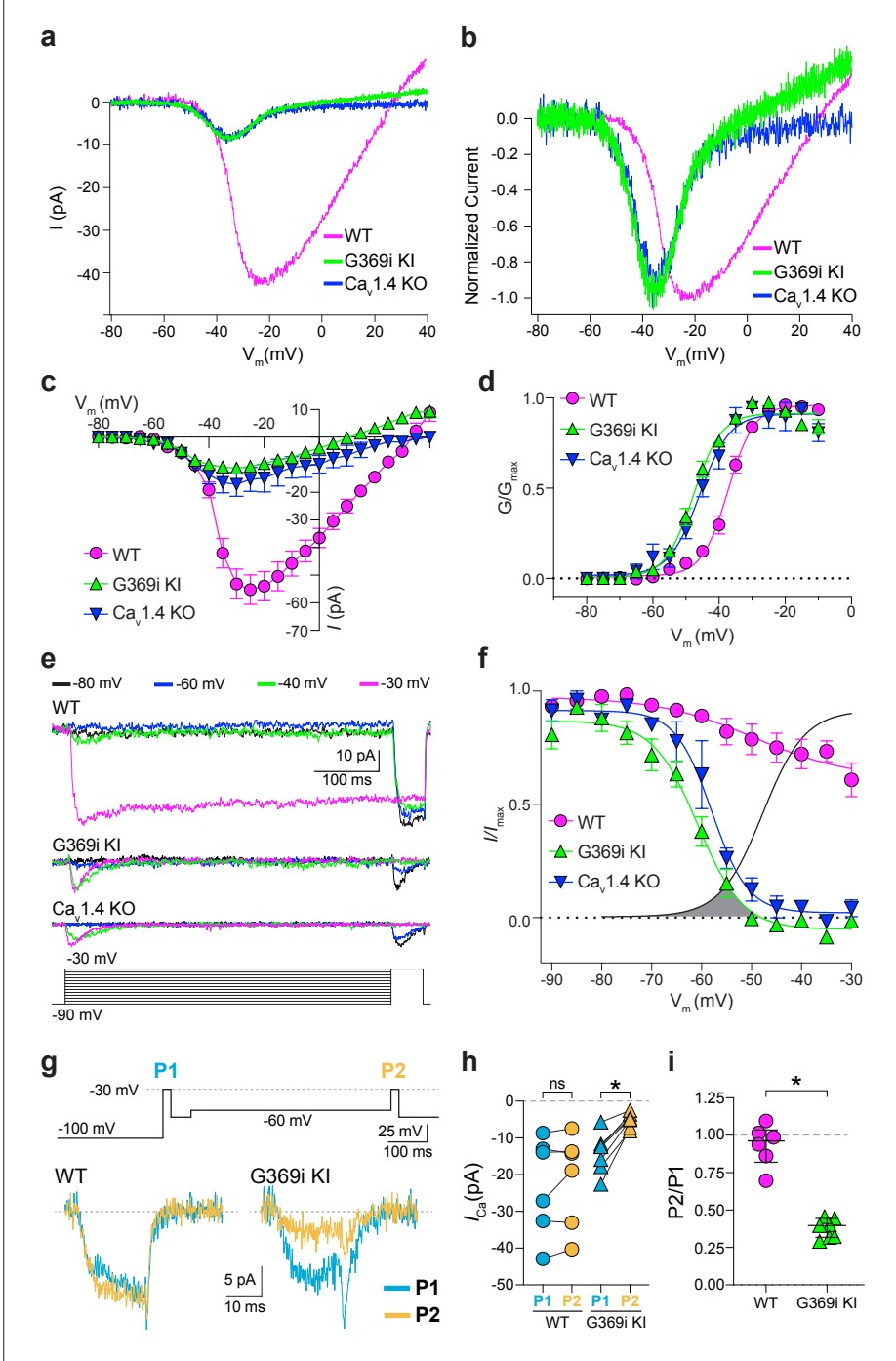

**Figure 2.** A low-voltage activated $I_{Ca}$ is present in cones of G369i KI and Ca$_V$1.4 knockout (KO) but absent in cones of wild-type (WT) mice. (**a**) Representative traces of $I_{Ca}$ evoked by voltage ramps. (**b**) $I_{Ca}$ traces from *a* normalized to their peak current amplitude to illustrate the hyperpolarizing shift in the I-V from cones of G369i KI and Ca$_V$1.4 KO mice. (**c,d**) I-V (**c**), and G-V (**d**) relationships for $I_{Ca}$ evoked by 50 ms, +5 mV increments from a holding voltage of –90 mV. V$_m$, test voltage. WT, n=13; G369i KI, n=9; Ca$_V$1.4 KO, n=3. (**e**) Representative $I_{Ca}$ traces and voltage protocol (bottom) for steady-state inactivation. $I_{Ca}$ was evoked by a conditioning pre-pulse from –90 mV to various voltages for 500ms followed by a test pulse to –30 mV for 50 ms. (**f**), Steady-state inactivation data from cones as recorded in *e*. I/I$_{max}$ represents the current amplitude of each –30 mV test pulse (I) normalized to current amplitude of the –30 mV test pulse preceded by a –90 mV conditioning pre-pulse (I$_{max}$) and was plotted against pre-pulse voltage. Shaded region indicates the window current for G369i KI cones. Line without symbols represents the G-V curve for G369i KI cones replotted from *d*. WT, n=8; G369i KI, n=8; Ca$_V$1.4 KO, n=3. In graphs *c,d,* and *f*, smooth

*Figure 2 continued on next page*

*Figure 2 continued*

lines represent Boltzmann fits, and symbols and bars represent mean ± SEM, respectively. (**g**) Representative $I_{Ca}$ traces from WT and G369i KI cones and voltage protocol (top). $I_{Ca}$ was evoked by a step from –100 mV to –30 mV before (**P1**) or after (**P2**) a 500ms step to –60 mV. (**h,i**) Graphs depicting peak amplitude of $I_{Ca}$ during P1 and P2 steps (**h**) and the ratio of the amplitudes of $I_{Ca}$ evoked by P2 and P1 pulses (P2/P1; **i**) from cones recorded as in *g*. WT, n=6; G369i KI, n=6, *p<0.05 by paired t-test.

The online version of this article includes the following source data and figure supplement(s) for figure 2:

**Source data 1.** Values obtained from recordings that were used for analyses in *Figure 2C, D, F, H, I* .

**Figure supplement 1.** Voltage ramps do not reveal $I_{Ca}$ in rods of G369i KI mice.

**Figure supplement 1—source data 1.** Values obtained from recordings that were used for analyses in *Figure 2— figure supplement 1C*.

In a previous study, a $Ca_v3$-like current was detected in patch clamp recordings of cone pedicles in WT mouse cones (*Davison et al., 2022*). However, in recordings of WT mouse cone somas, we did not observe a low-voltage activated component in the I-V or G-V curves that would be indicative of a $Ca_v3$ subtype (*Figure 2c and d*). To further test for a $Ca_v3$ contribution to $I_{Ca}$ in WT mouse cones, we used a double pulse voltage protocol where $I_{Ca}$ was evoked before (P1) and after (P2) a 500 ms step to –60 mV (*Figure 2g*). With this protocol, the inactivation of $Ca_v3$ channels by the –60 mV step should lead to a reduction in the amplitude of the P2 vs the P1 current. While such a reduction was clearly evident in cones of G369i KI mice (i.e. P2/P1 <1), there was no significant difference in the P2 and P1 currents in cones of WT mice (*Figure 2g–i*).

We also used antagonists selective for $Ca_v1$ and $Ca_v3$ channels (isradipine and ML 218, respectively). As expected, $I_{Ca}$ in WT cones was significantly but incompletely suppressed by isradipine (*Figure 3a and b*). The $V_h$ of the I-V was similar before (–36.5±0.8 mV) and after (–38.3±1.8 mV) perfusion of isradipine (t(3,4)=0.90, p=0.43 by paired t-test), suggesting that the residual $I_{Ca}$ after exposure to isradipine arose from unblocked $Ca_v1.4$ rather than $Ca_v3$ channels. While ML 218 modestly inhibited $I_{Ca}$ in some WT cones, the effect was not statistically significant (*Figure 3c and d*) and could be attributed to weak activity on $Ca_v1.4$. In transfected HEK293T cells, ML 218 modestly inhibited $I_{Ca}$ and caused a negative shift in the $V_h$ for $Ca_v1.4$, in contrast to its strong inhibition of $Ca_v3.2$ (*Figure 3— figure supplement 1*). Like its effects on $Ca_v3.2$, ML 218 nearly abolished $I_{Ca}$ in cones of G369i KI mice whereas isradipine had little effect (*Figure 3a–d*).

Finally, we recorded from ground squirrel and macaque cones, where the large amplitude of $I_{Ca}$ facilitates pharmacological and biophysical analyses. Consistent with its actions on $Ca_v1.4$ in WT mouse cones (*Figure 3b*) and transfected HEK293T cells (*Figure 3—figure supplement 1*), ML 218 caused an insignificant inhibition of peak $I_{Ca}$ (–7.0 ± 20.8%, n=6 cones) as well as a negative shift in the voltage-dependence of activation in ground squirrel cones ($\Delta V_{1/2}$ = -1.16 ± 0.94 mV, n=6, mean ± SD, p=0.029, t-test; *Figure 4a and b*). By contrast, the application of isradipine dramatically suppressed $I_{Ca}$ in cones in various regions of the ground squirrel retina (*Figure 4c and d*). Subsequent application of ML 218 did not reveal a $Ca_v3$-like current: the I-V and G-V of the isradipine + ML 218-sensitive current were shifted in the positive rather than the negative direction relative to the isradipine-sensitive current (*Figure 4c–g*). This result is consistent with the time- and voltage-dependent block of $Ca_v1$ channels by dihydropyridine antagonists such as isradipine (*Koschak et al., 2003*). As a positive control, we confirmed that ML 218 blocked a prominent $Ca_v3$-type current in ground squirrel type 3 a OFF cone bipolar cells (*Figure 4h and i*). In recordings of macaque cone pedicles, we compared the properties of $I_{Ca}$ using a holding voltage of –90 mV or –50 mV, the latter of which should inactivate $Ca_v3$ channels (*Figure 5a*). There was no difference in the I-V or G-V relationships obtained at these holding voltages (*Figure 5b–e*). As in WT mouse cones (*Figure 2g–i*), there was also no difference in $I_{Ca}$ recorded before and after an inactivating pulse to –60 mV (*Figure 5f and g*). We conclude that $Ca_v3$ channels do not normally contribute to $I_{Ca}$ in cones of WT mice, ground squirrels, and macaques.

To test whether the appearance of the $Ca_v3$-mediated $I_{Ca}$ in G369i KI cones could result from an increase in the expression of a specific $Ca_v3$ subtype (i.e. $Ca_v3.1$, $Ca_v3.2$, or $Ca_v3.3$), we initially tried labeling with commercially available antibodies against $Ca_v3.2$. However, the data were unreliable since these antibodies produced a similar labeling pattern in $Ca_v3.2$ KO retinal tissue (data not shown). As an alternative, we performed single-cell RNA sequencing on cells isolated from the retina of WT and G369i KI mice (*Figure 3—figure supplement 2*). We sequenced 28,782 cells and, with

**Table 1.** Comparison of parameters from electrophysiological recordings of cones.

| | $C_M$ (pF) | p-value | n | $R_M$ (MΩ) | p-value | n |
|---|---|---|---|---|---|---|
| WT | 4.88 (3.86, 5.01) | -- | 11 | 3.57 (2.81, 4.65) | -- | 11 |
| G369i KI | 4.08 (3.88, 4.34) | 0.73* | 9 | 5.06 (4.04, 8.24) | 0.04* | 9 |
| Ca$_v$1.4 KO | 3.31 (3.00, 3.76) | 0.001*, 0.04$^b$ | 6 | 8.21 (7.26, 9.06) | 0.001*, 0.47$^b$ | 6 |
| | | | | | | |
| G-V | $V_h$ (mV) | p-value | n | $k$ | p-value | n |
| WT | −37.54 (−38.91, −35.13) | -- | 13 | 2.75 (2.47, 4.32) | | 13 |
| G369i KI | −47.59 (−50.07, −46.72) | 0.0001* | 9 | 2.99 (2.68, 4.16) | 0.99* | 9 |
| Ca$_v$1.4 KO | −44.98 (−50.15, −43.27) | 0.085$^a$, 0.99$^b$ | 3 | 4.33 (3.73, 5.22) | 0.34*, 0.49$^†$ | 3 |
| | | | | | | |
| Steady-state inactivation | $V_h$ (mV) | p-value | n | $k$ | p-value | n |
| WT | −49.06 (−57.95, −41.82) | -- | 8 | −7.25 (−12.30, −4.97) | -- | 8 |
| G369i KI | −59.40 (−62.73, −58.01) | 0.01* | 8 | −3.81 (−5.23, −2.46) | 0.12$^a$ | 8 |
| Ca$_v$1.4 KO | −56.62 (−61.65, −56.56) | 0.88*, 0.88$^†$ | 3 | −3.67 (−5.17, −1.19) | 0.14$^a$, 0.99$^b$ | 3 |
| | | | | | | |
| | Tau activation | p-value | n | Tau deactivation | p-value | n |
| WT | 2.04 (1.26, 2.71) | -- | 11 | 0.88 (0.62, 2.09) | -- | 6 |
| G369i KI | 3.32 (2.99, 4.06) | 0.001* | 9 | 3.40 (2.62, 4.34) | 0.004* | 6 |

Values represent median (25th, 75th quartiles). Vh and k were determined from Boltzmann fits of the G-V and steady-state inactivation curves. Time constant (tau) for activation was obtained from exponential fit of the rising phase of ICa evoked by a 50-ms test pulse to a voltage near the peak of the I-V. Tau deactivation was determined from exponential fit of the decay of the tail current evoked by repolarization to -90 mV from +20 mV. CM, membrane capacitance; RM, input resistance. P-values were determined by Kruskal Wallis test.

*relative to WT.

$^†$relative to G369i KI.

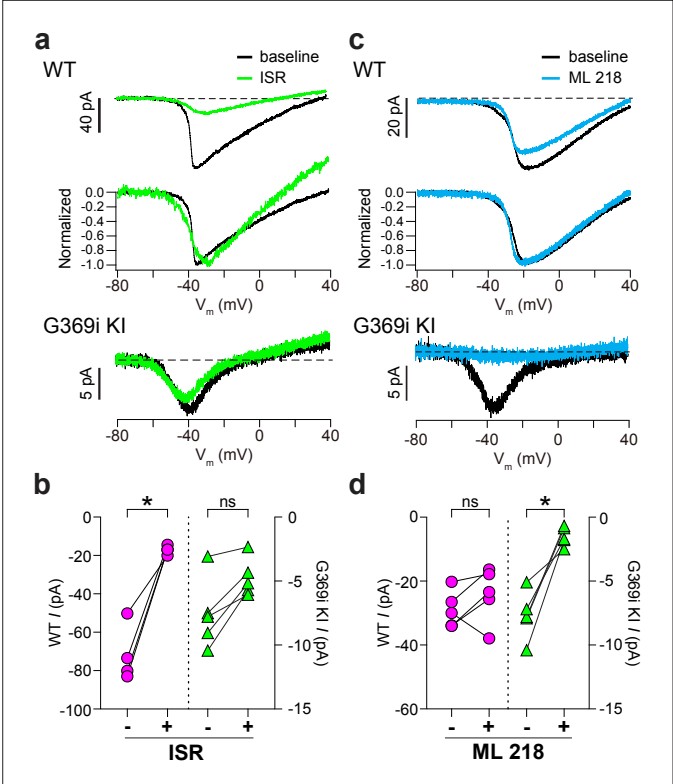

**Figure 3.** Pharmacological characterization of $I_{Ca}$ in cones of wild-type (WT) and G369i KI mice. (**a**) Representative traces for $I_{Ca}$ evoked by voltage ramps in cones of retinas from WT (top) and G369i KI (bottom) mice before (baseline) and after exposure to isradipine (ISR, 1 μM). Middle panel, WT $I_{Ca}$ from voltage ramps in the top panel were normalized to their peak $I_{Ca}$ to clarify the similar properties of $I_{Ca}$ before and after ISR exposure. (**b**) Peak $I_{Ca}$ before (-) and during (+) ISR exposure from cones as recorded in *a*. WT, n=4; G369i KI, n=5; *p<0.05 by paired t-test. (**c**) Representative traces for $I_{Ca}$ evoked by voltage ramps in cones of retinas from WT or G369i KI mice before (baseline) and after exposure to ML 218 (5 μM). Middle panel, WT $I_{Ca}$ from voltage ramps in the top panel were normalized to their peak $I_{Ca}$ to clarify the similar properties of $I_{Ca}$ before and after ML 218 exposure. (**d**) Peak $I_{Ca}$ before (-) and during (+) ML 218 exposure. WT, n=5; G369i KI, n=5; *p<0.05 by paired t-test.

The online version of this article includes the following source data and figure supplement(s) for figure 3:

**Source data 1.** Values obtained from recordings that were used for analyses in *Figure 3b and d*.

**Figure supplement 1.** Effect of ML218 on Ca$_v$1.4 and Ca$_v$3.2 in HEK293T cells.

**Figure supplement 1—source data 1.** Traces corresponding to $I_{Ca}$ are obtained in *Figure 3—figure supplement 1a, e*, and values obtained from recordings that were used from analyses in *Figure 3—figure supplement 1b–d, f-h*.

**Figure supplement 2.** Violin plots showing expression levels of transcripts at the single cell level in the retina of wild-type (WT) and G369i KI mice.

unsupervised clustering at resolution = 0.5, we grouped the cells into 20 clusters. Clusters 8 and 16 were identified as cones based on the expression of cone marker genes (e.g. *Opn1sw*, *Opn1mw*, *Arr3*, *Gnat2*). Consistent with previous studies (*Davison et al., 2022*; *Williams et al., 2022*), *Cacna1h* encoding the Ca$_v$3.2 subtype was the major Ca$_v$3 transcript expressed in WT mouse cones. There was no significant difference in *Cacna1h* expression between WT and G369i KI cones according to this analysis (*Figure 3—figure supplement 2*). Thus, the enhanced activity of Ca$_v$3 channels in cones of G369i KI mice results from a mechanism other than increased transcription of *Cacna1h*.

## Cone synapses are enlarged with some errors in postsynaptic wiring in G369i KI mice

The replacement of Ca$_v$1.4 with Ca$_v$3 as the primary conduit for Ca$^{2+}$ influx in G369i KI cones allowed us to dissect the importance of Ca$_v$1.4 Ca$^{2+}$ signals for the molecular and structural organization of

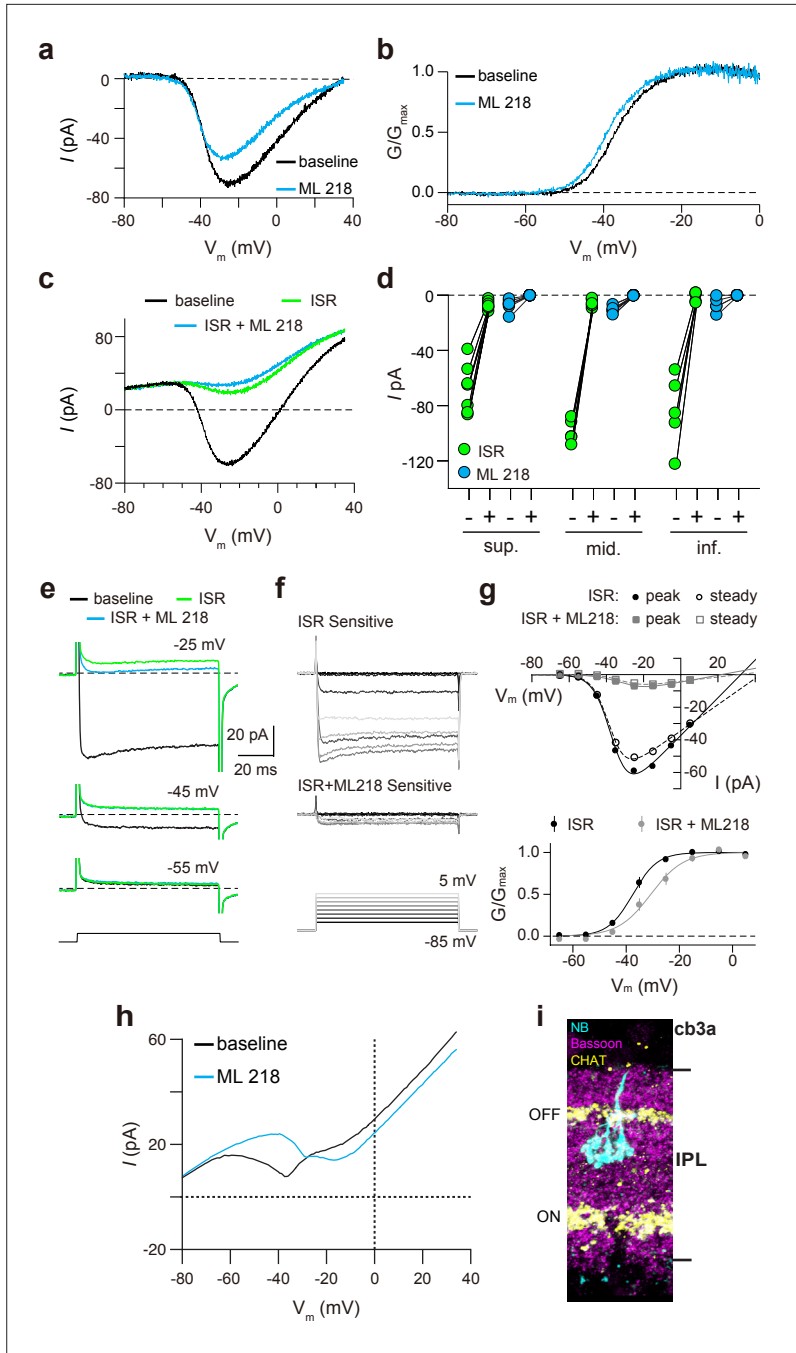

**Figure 4.** Pharmacological characterization of $I_{Ca}$ in ground squirrel cones. (**a**) Representative traces corresponding to baseline-corrected $I_{Ca}$ evoked by voltage ramps before (baseline) and during the application of ML 218. (**b**) G-V relationship of $I_{Ca}$ in *a*. (**c**) Representative traces corresponding to $I_{Ca}$ evoked by voltage ramps in control (baseline) and in ISR (2 µM) alone or in ISR + ML 218 (5 µM). (**d**) Peak $I_{Ca}$ from cones including the record from *c* before (-) and during (+) ISR or ISR +ML 218 block. The changes during the addition of ML218 were small but significant (sup., n=7, ISR: p<0.0001, ML218: p=0.0047; mid., n=5, ISR: p<0.0001, ML218: p=0.0001; inf., n=5, ISR: p=0.0019, ML218: p=0.0460; two-tailed t-test) and were likely due to a further non-specific, time-dependent reduction in the Ca$_v$1 current. (**e**) ICa was evoked by steps from –85 mV to voltages between –65 and –5 mV in increments of +10 mV (protocol shown below the current traces in *f*). Representative traces are shown in control, ISR, or ISR +ML 218. Current traces during steps to –55 and –45 mV lacked both transient and ML 218-senstive components. Dashed lines indicate zero current. (**f**) Traces show the $I_{Ca}$ sensitive to ISR (top) and ISR + ML218 (middle). The $I_{Ca}$ recorded in ISR was subtracted from the baseline $I_{Ca}$ (top). $I_{Ca}$ recorded in ISR + ML218 was subtracted from $I_{Ca}$ recorded in ISR (middle). (**g**) *Top*, I-V relationship of peak and steady-state $I_{Ca}$ from *f*. *Bottom*, the G-V relationship of the $I_{Ca}$ is

*Figure 4 continued on next page*

*Figure 4 continued*

blocked by ISR and ISR + ML218. Smooth lines represent Boltzmann fits. Symbols and bars represent mean ± SEM (n=4 cones). Due to a negative shift in the activation properties caused by ML 218 on a residual $Ca_v1$ current that is assumed to remain at the end of the experiment, the G-V curve of the $I_{Ca}$ isolated through subtraction by applying ISR + ML 281 underwent a statistically insignificant shift to the right. Data presented in *e-g* are from the same cone. (**h**) Representative $ICa$ traces from voltage ramps in an OFF cb3a bipolar cell before and during the application of ML 218. $V_{1/2}$ was shifted to the right by 13.1 ± 4.3 mV (n=3 cb3a cells, mean ± SD) consistent with the block of a $Ca_v3$-like conductance. In cb3b OFF bipolar cells, ML 218 produced only a slight leftward shift in an $I_{Ca}$ that had a more depolarized activation range, consistent with the exclusive expression a $Ca_v1$-type current ($\Delta V_{1/2}$ = -1.9 ± 0.4 mV, n=3 cells; data not shown). (**i**) Neurobiotin fill shows cb3a axon stratification and morphology in the retina labeled with antibodies against bassoon and choline acetyltransferase (CHAT) to label the OFF and ON sublamina of the IPL.

The online version of this article includes the following source data for figure 4:

**Source data 1.** Representative traces and values were obtained from recordings that were used for analyses in *Figure 4*.

the cone synapse. To this end, we first analyzed WT and G369i KI cone synapses by immunofluorescence and confocal microscopy. Presynaptic proteins such as bassoon and members of the postsynaptic signaling complex in depolarizing (ON) cone bipolar cells (GPR179, mGluR6, and TRPM1 *Martemyanov and Sampath, 2017*) were enriched near cone ribbons in G369i KI mice as in WT mice (*Figure 6a*). Compared to WT mice, the labeled structures in G369i KI mice occupied a larger volume (*Figure 6b and c*) but were of lower intensity (*Figure 6d*), suggesting an increase in the spread rather than in the levels of synaptic proteins. Unlike in WT mice, the volume of presynaptic and postsynaptic proteins increased linearly with the volume of the pedicle in G369i KI mice (*Figure 6e–i*). These results show that the molecular determinants of the cone synapse can assemble in G369i KI mice but change in their sub-synaptic distribution in ways that correlate with the expansion of the cone pedicle.

The cone synapse is structurally complex, with the dendritic tips of two horizontal cells and an intervening ON cone bipolar cell invaginating deeply into the pedicle near the ribbon (*Haverkamp et al., 2000*). To test how the switch in $Ca_v$ subtypes might affect this arrangement of postsynaptic partners, we generated 3D reconstructions of cone synapses by serial block-face scanning electron microscopy (SBFSEM; *Figure 7*). As with the enlarged synaptic contacts (*Figure 6*), structural modifications in G369i KI pedicles were evident. Compared to WT, ribbons in G369i KI pedicles appeared disorganized and were often parallel rather than perpendicular to the presynaptic membrane (*Figure 7a–c*). Consistent with our confocal analyses (*Figure 1*), G369i KI cone pedicles extended telodendria in multiple directions rather than just apically (*Figure 7a*). In addition, a slightly larger fraction of synaptic sites in the G369i KI pedicle (35% vs 14% in WT) formed incorrect postsynaptic partnerships including with glia (*Figure 7b–d'*; *Table 2*). Nevertheless, the number of ribbons is normal in G369i KI cones, and about half of the ribbons make invaginating contacts with the appropriate cell types (i.e. both horizontal cells and cone bipolar cells; *Table 2*). On average, there were more vesicles associated with ribbons in cones of G369i KI mice (245 per ribbon) than of WT mice (178 per ribbon; *Table 2*), but the difference did not reach statistical significance. Together with our immunofluorescence data, these results indicate that cone synapses undergo relatively subtle morphological changes in G369i KI mice, perhaps in response to the loss of $Ca_v1.4$ $Ca^{2+}$ signals.

## Postsynaptic responses to light are present in horizontal cells of G369i KI mice but not $Ca_v1.4$ KO mice

Compared to $Ca_v1.4$ KO mice, the relative preservation of cone synapses in G369i KI mice allowed the unique opportunity to test whether a $Ca_v$ subtype other than $Ca_v1.4$ could support synaptic release. To this end, we analyzed synaptic transmission between cones and horizontal cells (HCs). In the darkness, glutamate released from cones depolarizes horizontal cells via activation of α-amino-3-hydroxy-5-methyl-4-isoxazolepropionic (AMPA)/kainate receptors (*Hack et al., 2001*; *Brandstätter et al., 1997*). The resulting excitatory postsynaptic current (EPSC) undergoes a decline in response to light stimuli that hyperpolarize cones (*Feigenspan and Babai, 2015*). In WT HCs, a 1 s light flash ($\lambda$ =410 nm) inhibited the standing EPSC, which is reflected as an outward (hyperpolarizing 'ON') current (*Figure 8a*). Upon termination of the light, an inward (depolarizing 'OFF') current signaled the

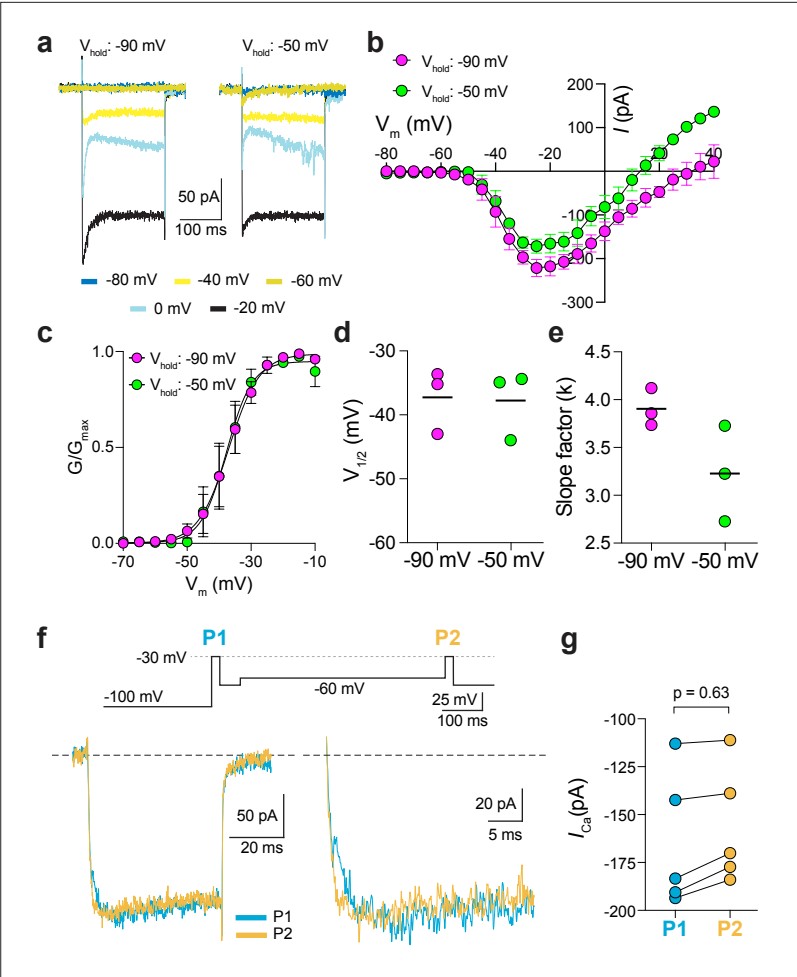

**Figure 5.** Characterization of $I_{Ca}$ in patch clamp recordings of cone pedicles of macaque retina. (**a-c**) Representative current traces (**a**) and corresponding I-V (**b**) and G-V (**c**) relationships for $ICa$ evoked by 200 ms steps from a holding voltage of –90 mV or –50 mV. n=3 cones. (**d, e**) $V_{1/2}$ (**d**) and slope factor (**e**) obtained from Boltzmann fit of data in *c*. p=0.50 in *d*; p=0.25 in *e* by Wilcoxon matched pairs signed rank test. (**f**) Voltage protocol (*top*) and representative $I_{Ca}$ traces (*bottom left*) from cones as recorded in **Figure 2g–i**. Bottom right, expanded view of the boxed region in the left traces. (**g**) Peak $I_{Ca}$ during P1 and P2 steps from cones recorded as in *f*. n=5 cones, p=0.62 by Wilcoxon matched pairs signed rank test.

The online version of this article includes the following source data for figure 5:

**Source data 1.** Values obtained from recordings that were used for analyses in **Figure 5b–e and g**.

resumption of the EPSC (**Figure 8a**). Both the ON and OFF responses in WT HCs increased with light intensity, reflecting the impact of luminance on the presynaptic membrane potential of cones and the subsequent change in glutamate release from their terminals (**Figure 8b**).

Consistent with the lack of mature ribbons and abnormal cone pedicles (**Figure 1**), HC light responses were negligible in $Ca_v1.4$ KO mice (**Figure 8a and b**). In contrast, the ON and OFF responses were present in G369i KI HCs although significantly lower in amplitude than in WT HCs (**Figure 8a and b**). As expected, the EPSC in darkness and the light responses were abolished by the AMPA/kainate receptor antagonist, DNQX, in both WT and G369i KI HCs (**Figure 8c and d**). In addition to its smaller amplitude, the transient nature of the ON response in G369i KI HCs suggested inadequate cessation of cone glutamate release by light (**Figure 8b**). Slow deactivation of $Ca_v3$ channels and/or their activation at negative voltages (**Perez-Reyes, 2003**) could give rise to $Ca^{2+}$ signals that support release following light-induced hyperpolarization of G369i KI cones.

To test if the diminished HC light responses correlated with lower presynaptic $Ca^{2+}$ signals in G369i KI cones, we performed 2-photon imaging of vertical slices prepared from the whole retina that was

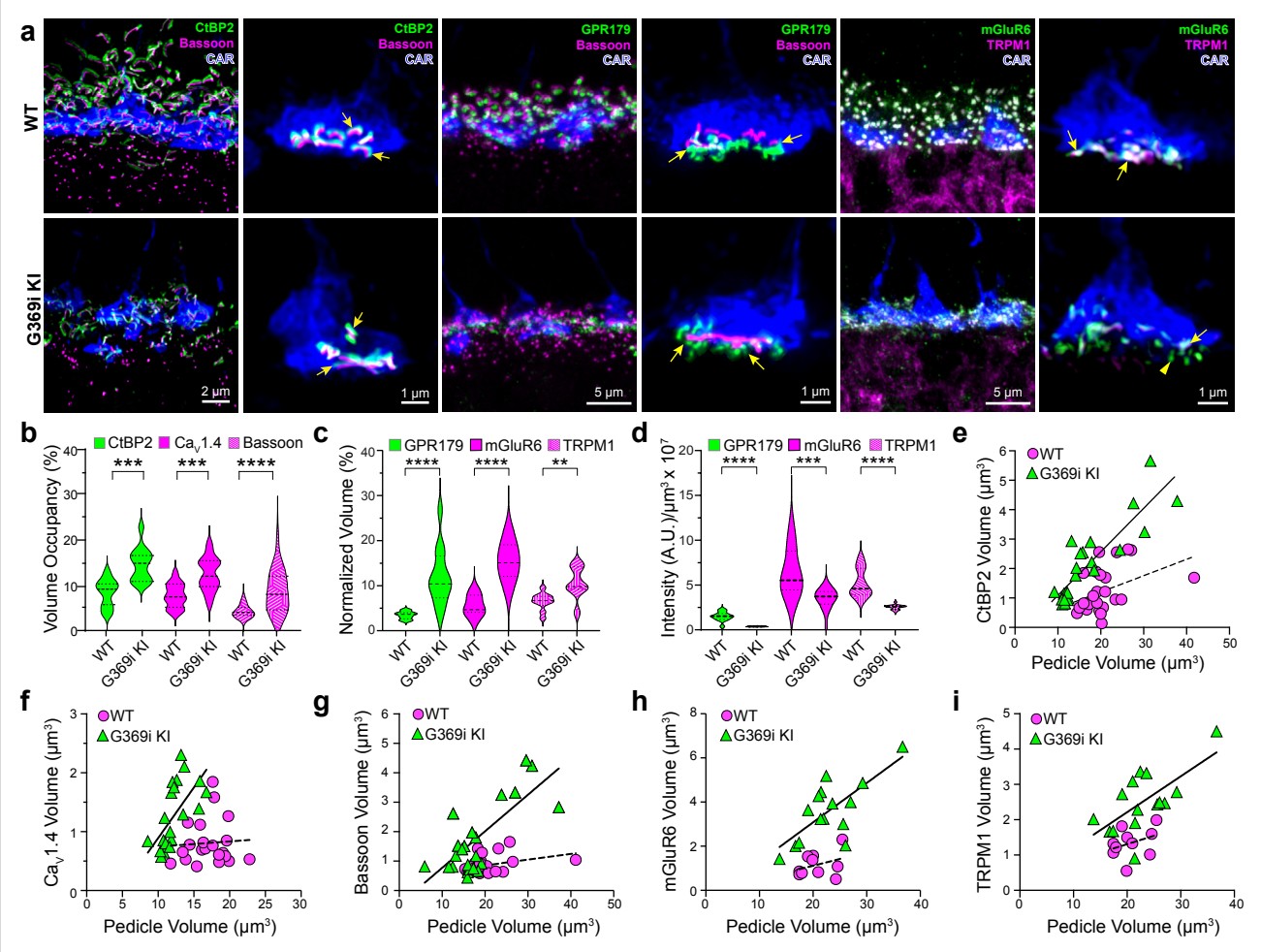

**Figure 6.** Immunofluorescence characterization of cone synapses in wild-type (WT) and G369i KI mice. (**a**) Confocal images of the outer plexiform layer (OPL) of WT and G369i KI mice labeled with antibodies against cone arrestin (CAR) and proteins that are presynaptic (CtBP2, bassoon) or postsynaptic (GPR179, TRPM1, mGluR6). Every other panel shows high-magnification, deconvolved images of single pedicles labeled with cone arrestin (rod spherule-associated signals were removed for clarity). Arrows indicate ribbon synapses, which appear enlarged in the G369i KI pedicles. (**b**) Violin plot representing the volume of presynaptic protein labeling normalized to the volume of their respective CAR-labeled pedicles (volume occupancy). (**c**) Violin plot representing the volume of postsynaptic protein labeling normalized to the volume of their respective CAR-labeled pedicles. (**d**) Violin plot representing fluorescence intensity of postsynaptic proteins. For *b-d*, **p<0.01, ***p<0.001, ****p<0.0001, unpaired t-tests.(**e–i**), Dependence of synapse size on pedicle size. Volumes corresponding to labeling of CtBP2 (*e*: p=0.051, *r*=0.4 for WT; p<0.0001, *r*=0.88 for G369i KI), Ca$_v$1.4 (*f*: p=0.8, *r*=0.06 for WT; p=0.002, *r*=0.88 for G369i KI), bassoon (*g*: p=0.1, *r*=0.34 for WT; p<0.0001, *r*=0.75 for G369i KI), mGluR6 (*h*: p=0.32, *r*=0.35 for WT; p=0.002, *r*=0.73 for G369i KI) and TRPM1 (*i*: p=0.32, *r*=0.35 for WT; p=0.007, *r*=0.66 for G369i KI) are plotted against pedicle volume. Dashed and solid lines represent fits by linear regression for WT and G369i KI, respectively.

The online version of this article includes the following source data for figure 6:

**Source data 1.** Values obtained from images that were used for analyses in *Figure 6b–i*.

incubated with the Ca$^{2+}$ indicator Fluo3-AM and Alexa-568-conjugated peanut agglutinin (PNA) to demarcate regions of interest (ROIs) corresponding to cone pedicles. With this approach, Fluo3 fluorescence was detected only in photoreceptors and ganglion cells and not inner retinal cell-types (e.g. horizontal cells, bipolar cells, Mueller cell soma). Thus, Ca$^{2+}$ signals reported by Fluo3 fluorescence near PNA-labeling originated primarily from cones. In WT cones, depolarization by K$^+$ (50 mM) caused a sustained increase of Ca$^{2+}$ (ΔF/F$_0$=1.85 ± 0.1, n=11 terminals) that was significantly lower after the addition of isradipine (ΔF/F$_0$=1.44 ± 0.07, p=0.0015 by repeated measures one-way ANOVA with Tukey's multiple comparison posthoc test; F[1.608, 16.08]=55.17; p<0.0001; *Figure 9a and b*). The Ca$^{2+}$ signal remaining in the presence of isradipine could arise from unblocked Ca$_v$1.4 channels and/or Ca$^{2+}$ release from intracellular stores (*Križaj, 2012*). In G369i KI cones, the peak amplitude of the

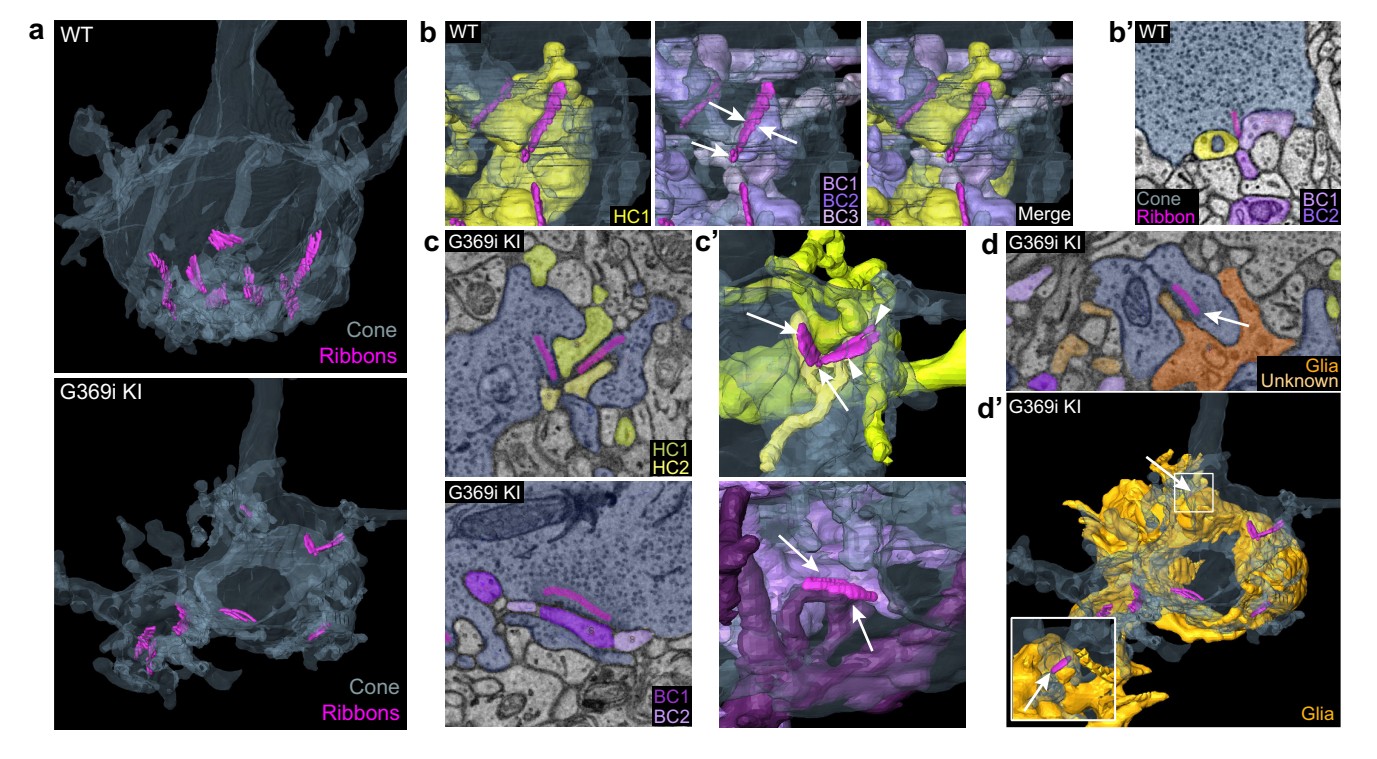

**Figure 7.** Three-dimensional reconstruction of cone synapses by SBFSEM. Reconstructions were made of wild-type (WT) and G369i KI pedicles (n=2 each). (**a**) 3D renderings showing ribbons (magenta) within cone pedicles (gray) from WT and G369i KI mice. (**b**) 3D renderings show ribbon sites in a WT cone pedicle contacting one horizontal (HC1; yellow) and three bipolar cells (BC1-3; purple). (**b'**) Raw image from *b* shows a single plane example of BC1-2 and HC1 contacting the ribbon site. (**c-d'**), Single plane raw (**c, d**) images and 3D reconstructions (**c', d'**) show ribbon sites within the G369i KI cone pedicle contacting in *c,c'*: horizontal cells (HC1-2) only (upper panels), CBCs (BC1-2) only (lower panels); and in *d,d'*: a glial cell (orange) and an unknown partner. The glial cell completely envelops the pedicle. Inset in *d'* shows glial-contacting ribbon site (arrow). In other panels, arrows indicate points of contact between ribbons and other postsynaptic elements.

K$^+$-evoked Ca$^{2+}$ signal ($\Delta F/F_0$=1.07 ± 0.02, n=14 terminals) was significantly lower than in WT cones (t(23)=8.74, p<0.0001 by unpaired t-test, n=14 terminals). The decrease in the Ca$^{2+}$ signal in the G369i KI cone pedicles during isradipine exposure paralleled that during the depolarization (*Figure 9a*), suggesting it was not a consequence of Ca$_v$1 blockade. The lower amplitude and more transient Ca$^{2+}$ signals mediated by Ca$_v$3 channels may be cleared more rapidly in G369i KI cones as compared to prolonged Ca$_v$1.4-initiated Ca$^{2+}$ signals in WT cones. Taken together, our results suggest that Ca$_v$3 channels nominally support Ca$^{2+}$ signals and synaptic transmission in cones of G369i KI mice.

## Light responses of bipolar cells and visual behavior are present in G369i KI but not Ca$_v$1.4 KO mice

While HCs play important roles in lateral inhibition of photoreceptor output, the vertical dissemination of visual information from cones to the inner retina is relayed by glutamate to ON and OFF cone bipolar cells (CBCs) which express mGluR6 and AMPA/kainate receptors, respectively. To test for the existence of cone-to-CBC transmission in G369i KI mice, we recorded electroretinograms (ERGs) under light-adapted conditions using Ca$_v$1.4 KO mice as a negative control (*Figure 10a–d*). In these recordings, the light-induced response of photoreceptors and the postsynaptic response of ON CBCs correspond to the a- and b-waves, respectively. Unlike in Ca$_v$1.4 KO mice, the a-waves of G369i KI mice resembled those in WT mice, which indicates that cones do not degenerate in this mouse strain (*Figure 10a and b*). While greatly reduced in amplitude, the b-wave was measurable in G369i KI mice and significantly larger than in Ca$_v$1.4 KO mice at the highest light intensities (F(14,104)=41, p<0.0001, post-hoc Tukey's p=0.0038 at 2.3 log cd·s/m$^2$; *Figure 10a and b*). We also recorded flicker ERGs using 10 Hz light stimuli that can isolate cone pathways involving both ON and OFF CBCs (*Nusinowitz*

**Table 2.** Comparison of parameters for cone synapse organization.

| | WT | | G369i KI | |
|---|---|---|---|---|
| | Cone 1 | Cone 2 | Cone 1 | Cone 2 |
| # of ribbons | 12 | 8 | 8 | 6 |
| Mean # synaptic vesicles per ribbon | 137.75 | 218.25 | 224 | 267.5 |
| Type of contact: % total (fraction of total) | | | | |
| *Invaginating* | 58.3 (7/12) | 87.5 (7/8) | 50 (4/8) | 50 (3/6) |
| *Non-invaginating* | 41.7 (5/12) | 12.5 (1/8) | 50 (4/8) | 50 (3/6) |
| Partner composition: % total (fraction of total) | | | | |
| *HC/BC* | 83.3 | 87.5 | 50.0 | 66.7 |
| *HC only* | 8.3 | 12.5 | 25.0 | 33.3 |
| *BC only* | 8.3 | 0 | 12.5 | 0 |
| *Glia only* | 0 | 0 | 12.5 | 0 |

Results represent analysis of n = 2 pedicles reconstructed by serial block face scanning electron microscopy in images obtained from WT or G369i KI mice (N = 1 animal each). The first row of vesicles encircling and immediately apposed to the ribbon were quantified for all the z- planes of the ribbon. Sites were counted as 'invaginating' if postsynaptic partners were enveloped on all sides by the cone terminal for at least a couple of consecutive image planes.

*et al., 2007*; *Vinberg et al., 2015*). In WT mice, flicker responses exhibited two peaks, one at a lower irradiance (–2 log cd·s/m²) and one at higher irradiance (0.5 log cd·s/m²; *Figure 10c and d*). The peak at the lower irradiance is attributed to responses in both rod and cone pathways, and the peak at a higher irradiance is attributed to responses exclusively in cone pathways (*Nusinowitz et al., 2007*; *Vinberg et al., 2015*). Flicker responses in G369i KI mice showed a similar non-monotonic relation as in WT mice but were significantly lower in amplitude (F(26,143)=51.18, p<0.0001, post-hoc Tukey's p=0.0237 at 2.5 log cd·s/m²; *Figure 10c and d*). The inverted flicker responses at higher illuminations in G369i KI mice were absent in Ca$_v$1.4 KO mice and may result from the hyperpolarizing contribution of cone-to-OFF CBC transmission (*Figure 10c*).

To test whether the cone synaptic responses in G369i KI mice enable vision-guided behavior, we used a swim test that assesses the ability of mice to identify a visible platform (*Bimonte-Nelson et al., 2015*). Under scotopic (dark) conditions, G369i KI and Ca$_v$1.4 KO mice took significantly longer to find the platform compared to WT mice (*Figure 10e and f*). These results are consistent with flicker response assays (*Figure 10c and d*) as well as the absence of $I_{Ca}$ in rods and rod-to-rod bipolar cell synaptic transmission in G369i KI mice (*Maddox et al., 2020*). Under photopic (light) conditions, G369i KI mice but not Ca$_v$1.4 KO mice performed as well as WT mice (*Figure 10e and f*). Thus, G369i KI mice have enough visual function under photopic conditions to quickly find the platform. Collectively, our results suggest that Ca$_v$3 channels can support cone synaptic responses that are sufficient for visual behavior in G369i KI but not Ca$_v$1.4 KO mice.

## Discussion

The nervous system has remarkable abilities to adapt to pathological perturbations in neuronal activity. In the retina, ablation or degeneration of photoreceptors triggers various postsynaptic mechanisms that maintain some level of visual function in rod or cone pathways. These include remodeling of bipolar cell dendrites and their synapses (*Beier et al., 2017*; *Shen et al., 2020*; *Leinonen et al., 2020*) as well as changes in the sensitivity of bipolar cells to photoreceptor input and inhibitory modulation

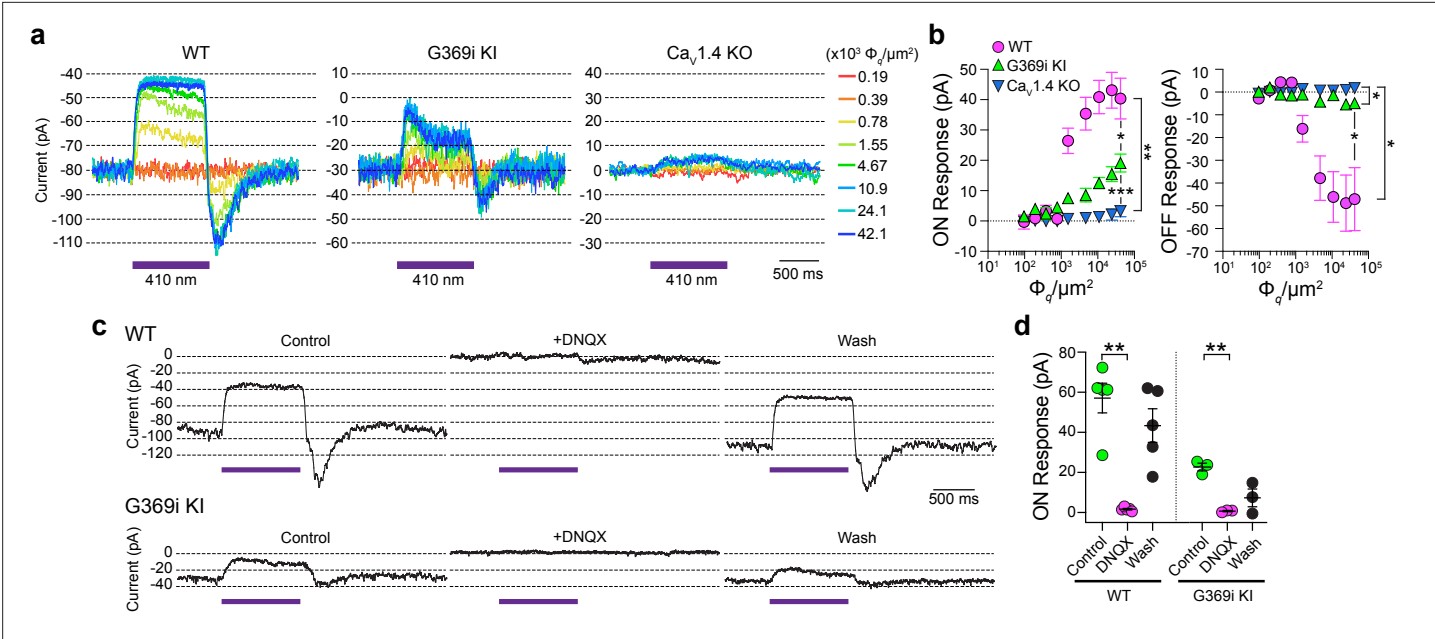

**Figure 8.** Patch clamp recordings of light responses of horizontal cells in ild-type (WT), G369i KI, and Ca$_v$1.4 knockout (KO) mice. (**a-b**) Representative traces from whole-cell patch clamp recordings of horizontal cells held at –70 mV (**a**) and quantified data (**b**) for currents evoked by 1 s light flashes ($\lambda$ =410 nm) plotted against light intensities. In *b*, peak current amplitudes during (ON Response) and after (OFF Response) the light stimuli were plotted against photon flux per µm$^2$ ($\Phi_q$/µm$^2$). Data represent mean ± SEM. WT, n=8; G369i KI, n=13; Ca$_v$1.4 KO, n=8. There was a significant difference in responses of WT, G369i KI, and Ca$_v$1.4 KO horizontal cells. For ON responses, F(16, 216)=22, p<0.0001 by two-way repeated measures ANOVA, Tukey's multiple comparisons post-hoc test. For OFF responses, F(16, 216)=13.61, p<0.0001 by two-way repeated measures ANOVA, Tukey's multiple comparisons post-hoc test. (**c–d**) Representative traces from horizontal cells held at –70 mV (**c**) and quantified data (**d**) for currents evoked by 1 s light flashes ($\lambda$ =410 nm, 1.2 × 10$^5$ $\Phi_q$/µm$^2$) before, during, and after washout of DNQX (20 µM). In *d*, symbols represent responses from individual cells, n=5 cells for WT and three cells for G369i KI, bars represent mean ± SEM. **p<0.01 by paired t-tests.

The online version of this article includes the following source data for figure 8:

**Source data 1.** Representative traces and values were obtained from recordings that were used for analyses in *Figure 8*.

---

(*Leinonen et al., 2020*; *Care et al., 2020*). To our knowledge, this study provides the first evidence for a presynaptic form of homeostatic plasticity that originates within photoreceptors. Using Ca$_v$1.4 KO and G369i KI mice, we identify the upregulation of a Ca$_v$3 conductance as a common response to Ca$_v$1.4 loss-of-function in cones. However, Ca$_v$3 channels can only compensate for Ca$_v$1.4 loss-of-function if the basic aspects of cone synapse structure are maintained through an expression of the G369i mutant channel. Thus, our results also highlight a crucial, non-conducting role for the Ca$_v$1.4 protein that allows cone synapses to function in the absence of Ca$_v$1.4 Ca$^{2+}$ signals.

## A non-canonical role for Ca$_v$1.4 in regulating cone synapse assembly

As shown for rod synapses (*Maddox et al., 2020*), ribbons and other components of the pre- and post-synaptic complex assemble at cone synapses in G369i KI mice (*Figures 1, 6 and 7*). Ca$_v$3 Ca$^{2+}$ signals are dispensable for this process since their presence in Ca$_v$1.4 KO cones is not accompanied by any semblance of ribbon synapses (*Figure 1b*). An intimate relationship between Ca$_v$1.4 and ribbons is supported by the colocalization of Ca$_v$1.4 and puncta resembling ribbon precursor spheres in the developing OPL (*Liu et al., 2013*). Ca$_v$1.4 interacts directly or indirectly with a variety of ribbon synapse-associated proteins including RIM (*Kiyonaka et al., 2007*), CAST/ERC2 (*Kiyonaka et al., 2012*), bassoon (*tom Dieck et al., 2005*), unc119 *Alpadi et al., 2008*; *Haeseleer, 2008*; *Haeseleer et al., 2004*, and a major component of the ribbon, RIBEYE (*Schmitz et al., 2000*). Ca$_v$1.4 could pioneer sites of ribbon assembly perhaps by serving as a docking or nucleation site for the active zone. At the same time, ribbon-associated proteins help cluster Ca$_v$1.4 near ribbons since knockout of these proteins in mice leads to fewer presynaptic Ca$_v$1.4 channels and in some cases shortened ribbons (*tom Dieck et al., 2012*). Regardless of the mechanism, our findings show that the formation of photoreceptor synaptic complexes does not require Ca$^{2+}$ influx through Ca$_v$1.4 channels.

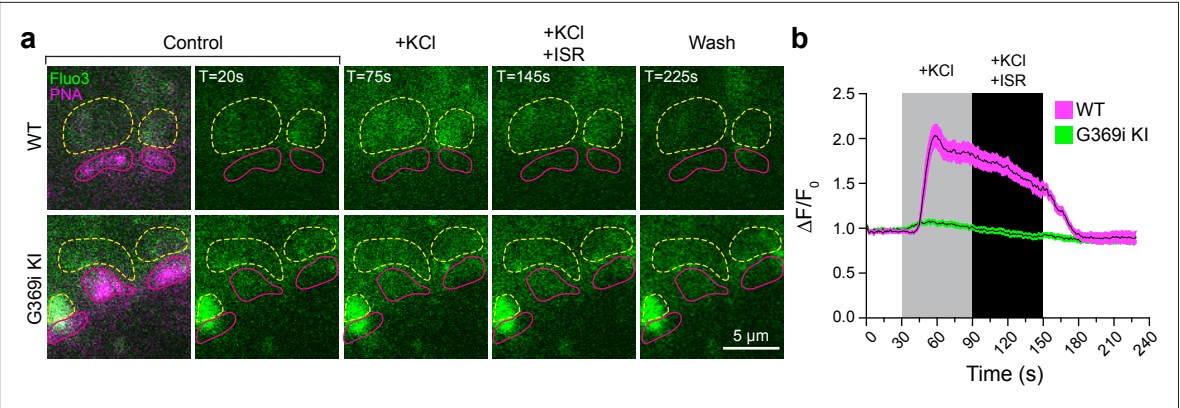

**Figure 9.** Ca²⁺ imaging of cone terminals in wild-type (WT) and G369i KI mice. Depolarization-evoked fluorescence changes (ΔF) were measured by two-photon imaging of Fluo3-filled cone pedicles labeled with PNA-Alexa 568. (**a**) Representative images at the indicated timepoints (**T**) before (control), during, and after (wash) bath application of KCl (50 mM) and KCl + ISR (10μ M). For clarity, PNA-Alexa 568 labeling was outlined (magenta, solid circles) and the actual labeling was only shown in the initial panel of images. Dashed, yellow circles represent ROIs for analyses. (**b**) Changes in cone pedicle Fluo3 fluorescence were normalized to the initial fluorescence intensity at time = 0 (ΔF/F₀). Solid lines and shaded areas represent the mean ± SEM, respectively. WT, n=11 pedicles; G369i KI, n=14 pedicles.

The online version of this article includes the following source data for figure 9:

**Source data 1.** Values obtained from time series ROIs that were used for analyses in *Figure 9b*.

## Functional Ca$_v$3 channels are present in cones in the absence of Ca$_v$1.4 Ca²⁺ signals

Our drop-seq analysis aligns with previous studies indicating that *Cacna1h* is expressed at extremely low levels compared to *Cacna1f* in WT mouse cones (*Figure 3—figure supplement 2*; *Williams et al., 2022*; *Macosko et al., 2015*). However, in contrast to recordings of WT mouse cone pedicles in a previous study (*Davison et al., 2022*), we found no evidence for Ca$_v$3-mediated currents in somatic recordings of cones in WT mice (*Figures 2 and 3*). We do not feel the discrepancy results from the different recording configurations given the electrically compact nature of cones. Moreover, recordings from macaque cone pedicles and from the ellipsoids of cylindrically shaped ground squirrel cones also did not reveal a Ca$_v$3-mediated current (*Figures 4 and 5*). We acknowledge that there are caveats of the pharmacological approach used in our study as well as the previous work (*Davison et al., 2022*). For example, dihydropyridine antagonists such as isradipine have relatively low affinity for some Ca$_v$1 subtypes, including Ca$_v$1.4 *Koschak et al., 2003*; *Koschak et al., 2001*, and have strongly voltage-dependent blocking properties. At a holding voltage of –90 mV, isradipine at 1 μM causes only ~80% inhibition of Ca$_v$1.4 with a greater block at depolarized voltages (*Koschak et al., 2003*), which agrees with our recordings of cones in WT mice (*Figure 3a*) and ground squirrels (*Figure 4b–e*). Moreover, in the low micromolar range, Ca$_v$3 blockers such as Z944 *Tringham et al., 2012* and ML 218 (*Figure 3—figure supplement 1*) have nonspecific effects on Ca$_v$1 current amplitude and activation voltage. Thus, Ca$_v$ channels mediating an $I_{Ca}$ that is suppressed by Ca$_v$3 blockers and spared by dihydropyridines at negative voltages may be mistakenly categorized as Ca$_v$3 subtypes. In our experiments, the biophysical properties of the isradipine-sensitive $I_{Ca}$ in cones of WT mice and ground squirrels resembled only those of Ca$_v$1, whereas those of the ML 218-sensitive $I_{Ca}$ in cones of G369i KI mice resembled only those of Ca$_v$3. Therefore, we favor the interpretation that Ca$_v$3 contributes to $I_{Ca}$ in mouse cones only upon silencing of Ca$_v$1.4 either by KO of the corresponding protein or by silencing its conductance.

Since *Cacna1h* expression was similar in WT and G369i KI cones (*Figure 3—figure supplement 2*), what causes the increased Ca$_v$3 channel activity in the latter? Among the Ca$_v$3 subtypes, Ca$_v$3.2 is uniquely sensitive to post-translational modifications that facilitate the opening of the channel and/or its cell surface trafficking (*Cai et al., 2021*). For example, Ca$_v$3.2 interacts with the deubiquitinating enzyme USP5 which enhances levels of Ca$_v$3.2 protein levels by suppressing its proteasomal degradation (*García-Caballero et al., 2014*). In addition, Ca$_v$3.2 is phosphorylated by calmodulin-dependent kinase II *Welsby et al., 2003* and cyclin-dependent kinase 5 (*Gomez et al., 2020*), which enhances

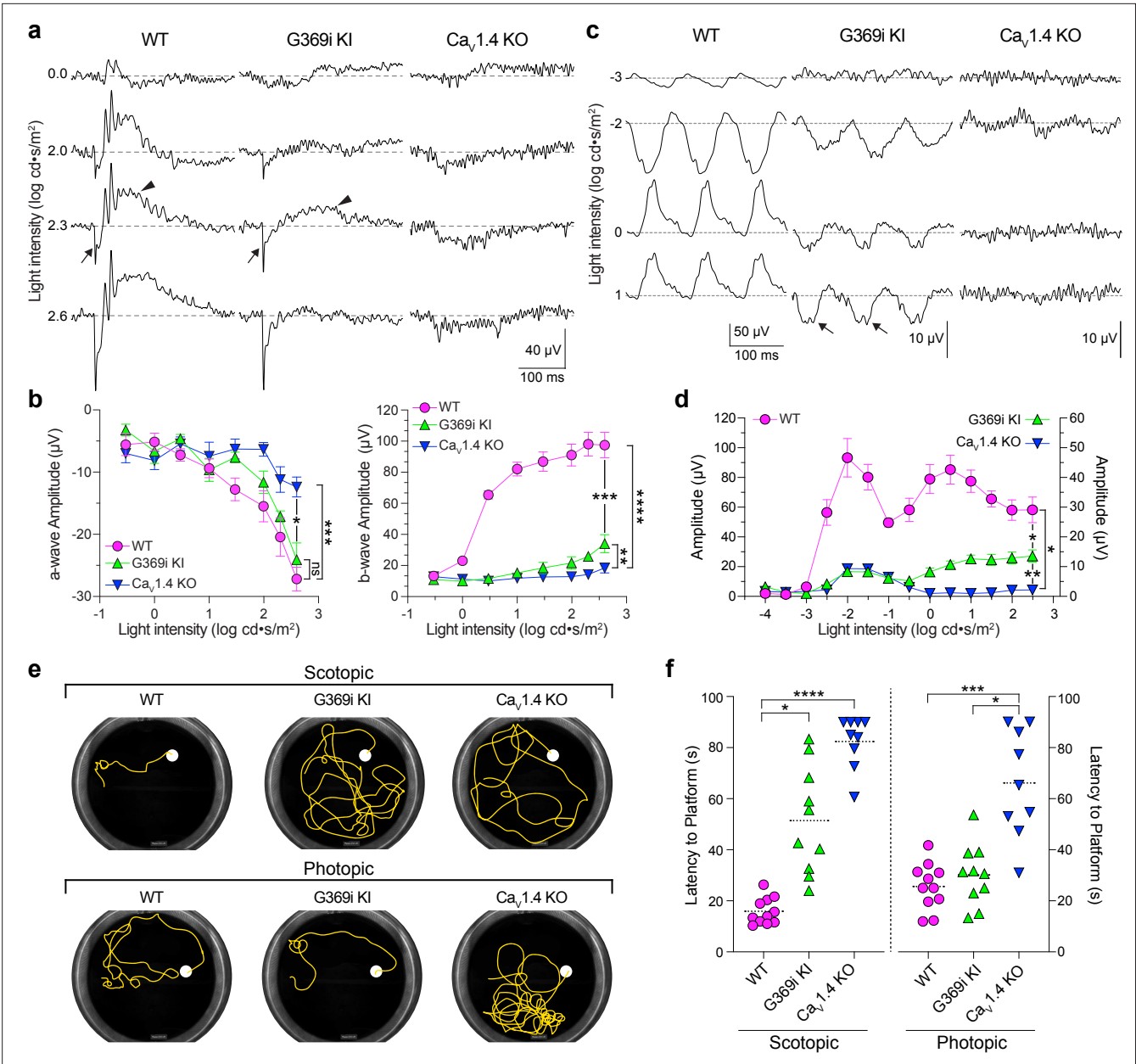

**Figure 10.** Characterization of electroretinograms (ERGs) and visual behavior in wild-type (WT), G369i KI, and Ca$_V$1.4 knockout (KO) mice. (**a**) Representative traces of photopic ERGs recorded in the presence of background green light (20 cd · s/m$^2$) in WT, G369i KI, and Ca$_V$1.4 KO mice. Flash intensities are shown on the left. Arrows and arrowheads depict the a- and b-waves, respectively. (**b**) a-wave (left), and b-wave (right) amplitudes are plotted against light intensity. Symbols and bars represent mean ± SEM. WT, n=7; G369i KI, n=5; Ca$_V$1.4 KO, n=6. *p<0.05; **p<0.01; ***p<0.001; ****p<0.0001; ns, not significant; Two-way ANOVA with Tukey's posthoc multiple comparisons. (**c, d**) Representative traces (**c**) and quantified data (**d**) for 10 Hz flicker responses evoked by white light flashes of increasing luminance (from −4–2 log cd· s/m$^2$). Arrows in *c* depict inverted waveform responses in G369i KI mice that are absent in Ca$_V$1.4 KO mice. Symbols and bars represent mean ± SEM. WT, n=7; G369i KI, n=5; Ca$_V$1.4 KO, n=6. *p<0.05; **p<0.01. Two-way ANOVA with Tukey's posthoc multiple comparisons. (**e**) Representative swim path traces of WT, G369i KI, and Ca$_V$1.4 KO mice from the visible platform swim tests performed in the dark (scotopic, upper panel) and light (photopic, lower panel). (**f**) Time required to reach the platform (latency to platform) was compared for each mouse strain. Symbols represent the average of the last 3 swim trials for each mouse of each genotype for both dark and light conditions. Dotted lines represent the mean. WT, n=11; G369i KI, n=10; Ca$_V$1.4 KO, n=9. *p<0.05; ***p<0.001; ****p<0.0001; Kruskal-Wallis one-way ANOVA with Dunn's post hoc analysis.

The online version of this article includes the following source data for figure 10:

**Source data 1.** Representative electroretinogram (ERG) traces and values obtained from ERGs and behavioral swim tests that were used for analyses in *Figure 10*.

channel activation and current density, respectively. Further interrogation of our drop-seq dataset could illuminate novel Ca$_v$3.2-regulatory pathways that are modified in G369i KI cones.

## Consequences of Ca$_v$1.4 loss of function and relevance for CSNB2

Despite having the same complement of proteins as in WT mice, cone synapses in G369i KI mice were enlarged and made errors in postsynaptic partner selection (*Figures 6 and 7*). While I-V curves indicate similar peak $I_{Ca}$ amplitudes in cones of WT and G369i KI mice near the membrane potential of cones in darkness (−45 to −50 mV *Ingram et al., 2019, Figure 2a and c*), the strong inactivation of Ca$_v$3 channels and/or reduced Ca$^{2+}$ release from intracellular stores likely account for the greatly diminished Ca$^{2+}$ signals in G369i KI pedicles (*Figure 9*). Paradoxically, reductions in presynaptic Ca$^{2+}$ in rod photoreceptors correlated with illumination-dependent shrinkage of ribbons in vivo and in vitro (*Dembla et al., 2020*; *Spiwoks-Becker et al., 2004*; *Regus-Leidig et al., 2010*). However, ribbon synapse size increases with low presynaptic Ca$^{2+}$ in zebrafish sensory hair cells *Sheets et al., 2012*. The mechanism involves decreased mitochondrial Ca$^{2+}$ uptake and an increase in the redox state of nicotinamide adenine dinucleotide (NAD$^+$/NADH ratio) (*Wong et al., 2019*). Ca$_v$3 Ca$^{2+}$ signals may decay too quickly in G369i KI cones (*Figure 9*) to enable mitochondrial Ca$^{2+}$ uptake or other mechanisms that trim ribbons.

An intriguing result was the linear increase in ribbon synapse size with pedicle volume in G369i KI but not in WT mice (*Figure 6e–i*), which could represent a form of homeostatic synaptic scaling as has been demonstrated at the neuromuscular junction (*Goel et al., 2019*; *Hong et al., 2020*). Similarly, the presence of some non-invaginating contacts with incorrect partner pairings in G369i KI mice (*Figure 7*, *Table 2*) could be an attempt to increase synaptic connectivity to compensate for weakened synaptic output. Consistent with this possibility, diminished presynaptic Ca$^{2+}$ signals and non-invaginating cone synapses also characterize the retina of mice lacking the extracellular α$_2$δ−4 subunit of Ca$_v$1.4 (*Kerov et al., 2018*). The cone synapses of G369i KI mice and α$_2$δ−4 KO mice are clearly functional given that photopic visual behavior is preserved in both mouse strains *Figure 10e and f*; *Kerov et al., 2018* and, in the case of G369i KI mice, HCs exhibit EPSCs whose amplitude decreases with light and AMPA/kainate receptor blockade (*Figure 8*). Defects in cone synaptic transmission that correlate with aberrant cone synapse morphology and connectivity also typify a mouse strain with a gain-of-function mutation in Ca$_v$1.4 linked to CSNB2 (*Zanetti et al., 2021*; *Knoflach et al., 2013*). Thus, balanced presynaptic Ca$^{2+}$ signals may be needed to encode the correct post-synaptic wiring and structure of the cone synapse.

In addition to structural abnormalities in cone synapses, the inability of Ca$_v$3 channels to support sustained presynaptic Ca$^{2+}$ entry likely contributed to the impaired light responses in HCs and CBCs in G369i KI mice (*Figures 8 and 10*). Due to their slow activation and strong inactivation (*Perez-Reyes et al., 2009*), Ca$_v$3 channels are not expected to fuel the Ca$^{2+}$ nanodomains that support fast and sustained components of release, both of which occur only at ribbon sites in cones (*Snellman et al., 2011*; *Choi et al., 2008*). Ca$_v$3 channels may also be located further from the ribbon than Ca$_v$1.4, thus lowering the efficiency of coupling to exocytosis. Regardless of the features that differentiate the contributions of Ca$_v$3 and Ca$_v$1.4 to cone synaptic release, our findings provide further support for the widely held view that Ca$_v$1 channels are well-suited to support transmission at ribbon-type synapses.

Based on extremely heterogeneous clinical presentations, CSNB2 manifests as a spectrum of visual disorders that originate from various mutations in *CACNA1F* (*Boycott et al., 2000*; *Bijveld et al., 2013*). Even though some CSNB2 mutant Ca$_v$1.4 channels may traffic normally to the plasma membrane, many of these mutations are expected to produce non-functional, non-conducting Ca$_v$1.4 channels (*Peloquin et al., 2007*; *Hoda et al., 2005*). Yet, the visual phenotypes of CSNB2 patients are not as severe as the complete blindness in Ca$_v$1.4 KO mice, which lack any Ca$_v$1.4 protein expression and exhibit no signs of visual behavior (*Figure 10e and f*; *Kerov et al., 2018*). Collectively, our results suggest that G369i KI mice accurately model Ca$_v$1.4 channelopathies in CSNB2 patients that are characterized by a greater impairment in rod than in cone pathways. This interpretation is supported by our findings that G369i KI mice exhibit horizontal cell responses to bright but not dim illumination (*Figure 8*), ERG responses under conditions of light adaptation (*Figure 10a–d*) but not dark-adaptation (*Maddox et al., 2020*), and visual behavior under photopic but not scotopic conditions (*Figure 10e and f*). Together with the enlargement of synaptic sites, modest levels of synaptic release from cones of G369i KI mice may support the nominal transmission of visual information

through cone pathways. G369i KI mice could also exhibit homeostatic alterations in the inner retina, which are known to support visual function when photoreceptor input is severely compromised (*Shen et al., 2020*; *Leinonen et al., 2020*; *Care et al., 2020*). Future studies of the retinal circuitry and visual behavior of G369i KI mice could identify compensatory pathways that are recruited upon Ca$_v$1.4 loss-of-function and how they might be targeted in novel therapies for CSNB2 and related disorders.

Values represent the median (25$^{th}$, 75$^{th}$ quartiles). $V_h$ and $k$ were determined from Boltzmann fits of the G-V and steady-state inactivation curves. Time constant (tau) for activation was obtained from the exponential fit of the rising phase of $I_{Ca}$ evoked by a 50 ms test pulse to a voltage near the peak of the I-V. Tau deactivation was determined from the exponential fit of the decay of the tail current evoked by repolarization to –90 mV from +20 mV. C$_M$, membrane capacitance; R$_M$, input resistance. p-values were determined by the Kruskal-Wallis test. [a], relative to WT. [b], relative to G369i KI.

Results represent the analysis of n=2 pedicles reconstructed by serial block face scanning electron microscopy in images obtained from WT or G369i KI mice (N=1 animal each). The first row of vesicles encircling and immediately apposed to the ribbon were quantified for all the z-planes of the ribbon. Sites were counted as 'invaginating' if postsynaptic partners were enveloped on all sides by the cone terminal for at least a couple of consecutive image planes.

# Methods

**Key resources table**

| Reagent type (species) or resource | Designation | Source or reference | Identifiers | Additional information |
|---|---|---|---|---|
| Cell line (*Homo sapiens*) | HEK293T | ATCC | CRL-3216 | CVCL_0063 |
| Strain, strain background (*Mus musculus*) | C57BL/6 J | Jackson Labs | 000664 | IMSR_JAX:000664 |
| Strain, strain background (*Mus musculus*) | G369i KI | Lee **Maddox et al., 2020** | N/A | NA |
| Strain, strain background (*Mus musculus*) | Ca$_v$1.4 KO | Jackson Labs | 017761 | IMSR_JAX:017761 |
| Strain, strain background (*Ictidomys tridecemlineatus*) | Ground Squirrel | TLC | N/A | NA |
| Biological sample (*Macaca mulatta*) | Retina | Seidemann | N/A | NA |
| Recombinant DNA reagent | CACNA1H | Genbank | AF051946.3 | Gift of E. Perez-Reyes |
| Recombinant DNA reagent | CACNA1F | Genbank | AF201304 | Gift of F. Haeseleer |
| Antibody | Bassoon (Mouse monoclonal) | ThermoFisher Scientific | Cat#: MA1-20689 RRID:AB_2066981 | 2 µg/mL |
| Antibody | Ca$_v$1.4 (Rabbit Polyclonal) | Amy Lee | Cat#: Ab167 RRID:AB_2650487 | 5 µg/mL |
| Antibody | Cone Arrestin (Rabbit Polyclonal) | Millipore | Cat#: AB15282 RRID:AB_1163387 | 5 µg/mL |
| Antibody | CtBP2 (Mouse monoclonal) | BD Biosciences | Cat#: 612044 RRID:AB_399431 | 1 µg/mL |
| Antibody | GPR179 (Mouse monoclonal) | Millipore | Cat#: MAB427 RRID:AB_2069582 | 2 µg/mL |
| Antibody | mGluR6 (Mouse monoclonal) | Melina Agosto | N/A | 1 µg/mL |
| Antibody | TRPM1 (Mouse monoclonal) | Melina Agosto | N/A | 1 µg/mL |
| Commercial assay, kit | Mix-n-Stain Antibody Labelling Kit | Biotium | 92238 | |
| Chemical compound, drug | AMES | Sigma, US Biological | Sigma: A1420; US Biological: A1372 | |
| Chemical compound, drug | BAPTA | Sigma-Aldrich | A4926 | |

*Continued on next page*

*Continued*

| Reagent type (species) or resource | Designation | Source or reference | Identifiers | Additional information |
|---|---|---|---|---|
| Chemical compound, drug | Antisedan | Zoetis US LLC | 10000449 | |
| Chemical compound, drug | TBF-TBOA | Tocris | 2532 | |
| Chemical compound, drug | DNQX | Abcam | ab120169 | |
| Chemical compound, drug | Isradipine | Sigma-Aldrich | I6658 | |
| Chemical compound, drug | ML218 | Tocris | 4507 | |
| Chemical compound, drug | Picrotoxin | Tocris | 1128 | |
| Chemical compound, drug | Strychnine | Sigma | S8753 | |
| Chemical compound, drug | Xylazine | Pivetal | 46066-0750-02 | |
| Chemical compound, drug | ZD7288 | Tocris | 1000 | |
| Software, algorithm | Amira | ThermoFisher Scientific | https://www.thermofisher.com/amira | |
| Software, algorithm | cellSens | Olympus | https://www.olympus-lifescience.com/en/software/cellsens | |
| Software, algorithm | Espion software | Diagnosys, Inc | https://info.diagnosysllc.com/software | |
| Software, algorithm | FLUOVIEW | Olympus | https://www.olympus-lifescience.com/en/laser-scanning/fv3000 | |
| Software, algorithm | GraphPad Prism | GraphPad | https://www.graphpad.com | |
| Software, algorithm | IGOR Pro | WaveMetrics | https://www.wavemetrics.com | |
| Software, algorithm | ImageJ | NIH | https://imagej.nih.gov/ij | |
| Software, algorithm | pClamp | Molecular Devices | https://www.moleculardevices.com | |
| Software, algorithm | Patchmaster | Harvard Bioscience, Inc | https://www.heka.com | |
| Software, algorithm | Scan Image | MBF Biosciences | https://docs.scanimage.org/index.html | |

## Animals

All animal experiments were performed in accordance with guidelines approved by the National Institutes of Health and the Institutional Animal Care and Use Committees at the University of Texas at Austin and Northwestern University were approved by the Institutional Animal Care and Use Committee (AUP-2023–00170). The G369i KI *Maddox et al., 2020* and Ca$_v$1.4 KO mouse strains were bred on the C57BL/6 J background strain for at least 10 generations. Adult male and female mice were used (6–12 weeks old), and aged-matched C57BL6/J mice were used as the control (WT) animals.

## Immunofluorescence

Mice between 6–8 weeks of age were anesthetized using isoflurane and euthanized by cervical dislocation. Eyes were enucleated and hemisected. The eye cups with retina were fixed on ice in 4% paraformaldehyde in 0.1 M phosphate buffer (PB) for 30 min. Fixed eye cups were then washed three times with 0.1 M PB containing 1% glycine followed by an infusion of 30% sucrose at 4 °C overnight. The eye cups were orientated along their dorsal-ventral axis and frozen in a 1:1 (wt/vol) mixture of Optimal Cutting Temperature compound and 30% sucrose in a dry ice/isopentane bath. Eye cups were cryosectioned at 20 µm on a Leica CM1850 cryostat (Leica Microsystems), mounted on Superfrost plus Micro Slides (VWR), dried for 5–10 min at 42 °C, and stored at –20 °C until used. Slides with mounted cryosections were warmed to room temperature, washed with 0.1 M PB for 30 min to remove the OCT/sucrose mixture and blocked with dilution solution (DS, 0.1 M PB/10%

goat serum/0.5% Triton-X100) for 15 min or overnight at room temperature. All remaining steps were carried out at room temperature. All primary antibodies and appropriate secondary antibodies were diluted in DS at concentrations specified in the Key Resources Table. Sections were incubated with primary antibodies for 1 hr or overnight and then washed five times with 0.1 M PB. Sections were then incubated with secondary antibodies for 30 min and then washed five times with 0.1 M PB. Trace 0.1 M PB was removed, and sections were then mounted with #1.5 H coverslips (ThorLabs) using ProLong Glass Antifade Mountant with or without NucBlue (Thermo Fisher Scientific).

For double labeling with other rabbit polyclonal antibodies, CAR antibodies were conjugated with the CF647 fluorophore (CAR-647) using the Mix-n-Stain antibody labeling kit according to the manufacturer's protocol (Biotium). Sections were processed first with rabbit polyclonal $Ca_v1.4$ or EAAT2 antibodies and corresponding secondary antibodies as described above. To prevent CAR-647 from binding to any available sites on the anti-rabbit secondary antibodies previously added to the sections, $Ca_v1.4$ or EAAT2 antibodies were readded to the sections and incubated for 30 min. After washing, the sections were incubated with CAR-647 for 1 hr, followed by wash steps and cover glass mounting.

Immunofluorescence in labeled retinal sections was visualized using an Olympus FV3000 confocal microscope (Tokyo, Japan) equipped with an UPlanApo 60 x oil HR objective (1.5 NA). Images were captured using the Olympus FLUOVIEW software package. Acquisition settings were optimized using a saturation mask to prevent signal saturation prior to collecting 16-bit. All confocal images presented are maximum z-projections. Images (256 × 256 pixels) used in analyses of synaptic proteins were collected using a 30 X optical zoom, 0.6 Airy disk aperture, and voxel size of 0.028 μm × 0.028 μm (X × Y × Z). Amira segmentation was used for generating 3D binary masks, and Amira 3D label analysis was used for quantification of immunofluorescent and masked images. Non-deconvolved images were used for all analyses. 3D binary masks of individual CAR-labeled pedicles were made by setting the threshold 1 standard deviation above the mean fluorescence intensity of each 3D *Figure . and 3D* binary masks of presynaptic labels (CtBP2, $Ca_v1.4$, and Bassoon) or postsynaptic labels (GPR179, mGluR6, and TRPM1) within or associated with the pedicle, respectively, were made by setting the threshold 3 standard deviations above the mean fluorescence intensity. To aid in presentation, high-mag images displayed in *Figures 1b and 6a* were deconvolved using cellSens software (Olympus), and any immunofluorescence corresponding to rod synaptic proteins was subtracted from the image. Experiments involving qualitative and/or quantitative analyses were performed independently at least three times, with one eye from per animal representing a biological replicate (three animals per genotype).

## Molecular biology and transfection

The cDNAs for $Ca_v1.4$ (GenBank: NM_019582), $β_{2X13}$ (GenBank: KJ789960), and $α_2δ–4$ (GenBank: NM_172364) were previously cloned into pcDNA3.1 *Lee et al., 2015*. The cDNA for $Ca_v3.2$ (GenBank: AF051946) was a gift from Dr. Edward Perez-Reyes, University of Virginia. All constructs were verified by DNA sequencing before use. Human embryonic kidney 293T (HEK293T) cells were cultured in Dulbecco's Modified Eagle's Medium with 10% FBS at 37 °C in 5% $CO_2$. At 70–80% confluence, the cells were co-transfected with cDNAs encoding human $Ca_v1.4$ (1.8μg) $β_{2X13}$ (0.6 μg), $α_2δ–4$ (0.6 μg), and enhanced GFP in pEGFP-C1 (Clonetech, 0.1 μg) or $Ca_v3.2$ (2μg) and pEGFP-C1 (0.1 μg) using FuGENE 6 transfection reagent according to the manufacturer's protocol. Cells treated with the transfection mixture were incubated at 37 °C for 24 hr, dissociated using Trypsin-EDTA, and replated at a low density to isolate single cells. Replated cells were then incubated at 30 °C or 37 °C for an additional 24 hr before beginning experiments.

## Solutions for patch clamp recordings

HEK293T extracellular recording solution contained the following (in mM): 140 Tris, 20 $CaCl_2$, 1 $MgCl_2$, pH 7.3 with methanesulfonic acid, and osmolarity 309 mOsm/kg. HEK293T internal recording solution contained the following: 140 NMDG, 10 HEPES, 2 $MgCl_2$, 2 Mg-ATP, 5 EGTA, pH 7.3 with methanesulfonic acid, osmolarity 358 mOsm/kg.

For recordings of $I_{Ca}$ in mouse retina, extracellular recording solution contained the following (in mM): 115 NaCl, 2.5 KCl, 22.5 $NaHCO_3$, 1.25 $NaH_2PO_4$, 2 $CaCl_2$, 1 $MgCl_2$, 5 HEPES, 5 CsCl, 5.5 Glucose, osmolarity 290 mOsm/kg. Mouse cone intracellular solution contained the following (in mM): 105 $CsMeSO_4$, 20 TEA-Cl, 1 $MgCl_2$, 11 HEPES, 10 EGTA, 4 Mg-ATP, 10 phosphocreatine, 0.3 Na-GTP, pH 7.4 with CsOH, osmolarity 300 mOsm/kg.

For recordings of $I_{Ca}$ in ground squirrel retina, extracellular recording solution contained the following (in mM): 10 HEPES, 85 NaCl, 3.1 KCl, 2.48 MgSO$_4$, 6 Glucose, 1 Na-succinate, 1 Na-malate, 1 Na-lactate, 1 Na-pyruvate, 2 CaCl$_2$, 25 NaHCO$_3$, and 20 TEA-Cl, osmolarity 285 ± 5 mOsm/kg. Intracellular solution contained the following (in mM): 80 CsCl, 10 BAPTA, 2 MgSO$_4$, 10 HEPES, 20 TEA-Cl, 5 Mg-ATP, and 0.5 Na-GTP, pH 7.35 with CsOH, osmolarity 285±5 mOsm/kg. For recordings of $I_{Aglu}$, extracellular recording solution contained (in mM): 125 NaCl, 3.0 KCl, 1.25 NaH$_2$PO$_4$, 25 NaHCO$_3$, 2 CaCl$_2$, 1 MgCl$_2$, 3 dextrose, 3 sodium pyruvate, 0.1 picrotoxin and 0.02 DNQX, and in indicated experiments 0.0013 TFB-TBOA. Intracellular $I_{Aglu}$ contained (in mM): 125 KSCN, 10 TEA-Cl, 10 HEPES, 1 CaCl$_2$, 2 MgCl$_2$, 0.3 Na-GTP, 4 Mg-ATP, 10 K$_2$ phosphocreatine, 0.02 ZD7288. Mouse and ground squirrel extracellular slice recording solutions were equilibrated with 5% CO$_2$/95% O$_2$ to a pH of ~7.5.

For recordings of light responses in horizontal cells of mouse retina, extracellular recording solution consisted of Ames' media supplemented with 100 U/mL penicillin, 0.1 mg/mL streptomycin, and 22.6 mM NaHCO$_3$, osmolarity 280±5 mOsm/kg. The intracellular recording solution contained the following (in mM): 135 K-Aspartate, 10 KCl, 10 HEPES, 5 EDTA, 0.5 CaCl$_2$, 1 Mg-ATP, 0.2 Na-GTP, pH 7.35 with KOH, osmolarity 305 ± 5 mOsm/kg.

**Table 3.** Pharmacological drugs and their concentrations used in electrophysiological recordings of cones.

| Compound | Concentration (µM) for mouse recordings | Concentration (µM) for ground squirrel recordings |
|---|---|---|
| Cadmium | 200 | N/A |
| DL-TBOA | N/A | 375 |
| DNQX | 20 | N/A |
| Isradipine | 1 | 2 |
| ML218 | 5 | 5 |
| Nickel | 100 | N/A |
| Picrotoxin | N/A | 50 |
| Strychnine | N/A | 10 |
| ZD7288 | N/A | 50 |
| Glutamate | 1000 | N/A |

## Patch clamp electrophysiology

Whole-cell voltage-clamp recordings of transfected HEK293T cells were performed 48–72 hr after transfection using an EPC-10 amplifier and Patchmaster software (HEKA Elektronik, Lambrecht, Germany). Patch pipette electrodes with a tip resistance between 4 and 6 MΩ were pulled from thin-walled borosilicate glass capillaries (World Precision Instruments, Sarasota, FL) using a P-97 Flaming/Brown Puller (Sutter Instruments, Novato, CA). A reference Ag/AgCl wire was placed into the culture dish mounted on an inverted Olympus IX70 microscope. Recordings were performed at room temperature. A pressurized perfusion pencil multi-barrel manifold controlled with Valve Bank II (AutoMate Scientific, Inc, Berkeley, CA) was used to deliver extracellular solutions. ML218 of different concentrations (0.5, 1, 5, 25, and 100 mM) was added to the extracellular solution on the day of the experiments. Series resistance was compensated up to 70%, and passive membrane leak subtraction was conducted using a P/−4 protocol. Whole-cell Ca$^{2+}$ currents ($I_{Ca}$) of transfected HEK293T cells were evoked for 50ms with incremental +5 mV steps from –80 mV to +65 mV. Current-Voltage (IV) data were fit with a single Boltzmann equation: $I_{Ca} = G_{max}(V_m-V_r)/(1+\exp[-(V_m-V_h)/k])$, where $G_{max}$ is the maximal conductance, $V_m$ is the test voltage, $V_r$ is the Ca$^{2+}$ reversal potential, $V_h$ is the membrane potential required to activate 50% of $G_{max}$, and k is the slope factor. Data were sampled at 100 kHz, filtered at 3 kHz, and analyzed using IgroPro (WaveMetrics).

Whole-cell voltage-clamp recordings of mouse cones, rods, and horizontal cells and macaque cone terminals were performed using an EPC-10 amplifier and Patchmaster software (HEKA). Patch pipette electrodes with a tip resistance between 10 and 14 MΩ for cones and 6–8 for horizontal cells were pulled from thick-walled borosilicate glass (1.5 mm outer diameter; 0.84 mm inner diameter; World Precision Instruments).

To prepare retinal slices, adult mice (6–8 weeks old) were anesthetized using isoflurane and euthanized by cervical dislocation. Eyes were enucleated, placed into cold Ames' media slicing solution, and hemisected. Following the removal of the vitreous, the eye cup was separated into dorsal and ventral halves using a scalpel. Ventral retina was isolated, molded into low-melt agarose, and mounted in a Leica VT1200s vibratome (Leica Biosystems). Mouse retina slicing solution was continuously bubbled

with 100% $O_2$ and contained the following: Ames' Medium with L-glutamine supplemented with (in mM) 15 NaCl, 10 HEPES, 10 U/mL penicillin, 0.1 mg/mL streptomycin, pH 7.4, osmolarity 300 mOsm/kg. Vertical (~200 µm) or horizontal (~160 µm) retinal slices were anchored in a recording chamber, placed onto a fixed stage, and positioned under an upright Olympus BX51WI microscope equipped with a 60 X water-immersion objective (1.0 NA), and superfused with extracellular solution (flow rate of ~1–2 ml/min) at room temperature. Slices were visualized using IR-DIC optics and an IR-2000 (Dage MTI, Michigan City, IN) or SciCam Pro (Scientifica, Uckfield, United Kingdom) CCD camera controlled by the IR-capture software package or µManager, respectively (*Edelstein et al., 2010*; *Edelstein et al., 2014*). Drugs used in these experiments were added to the mouse extracellular solution on the day of experiments at the concentration described in *Table 3*. A reference Ag/AgCl pellet electrode was placed directly into the recording chamber solution. Data from whole-cell recordings with a series resistance >20 MΩ were discarded.

Cone and rod somas were identified based on their morphology and location. Cone identity was confirmed by the whole-cell capacitance (~3–4 pF), which is larger than rod whole-cell capacitance (~0.7–1 pF). Unless otherwise indicated, cones and rods were held at –90 mV for 200ms followed by a ramp of +0.5 mV/ms to +40 mV. To determine the voltage activation of $I_{Ca}$, whole-cell $Ca^{2+}$ currents in cones were evoked for 50 ms with incremental +5 mV steps from –80 mV to +40 mV. The activation voltage of $I_{Ca}$ is reported as G/$G_{max}$, where G is the conductance at each test voltage and $G_{max}$ is the maximum peak conductance for each cone. Conductance was calculated using the equation $I_{Ca}$ = G($V_m$-$V_r$), where $V_r$ is +60 mV. To determine steady-state inactivation of $I_{Ca}$, currents in cones were evoked for 500 ms with incremental +5 mV steps from –90 mV and –30 mV followed by a final step to –30 mV for 50 ms after each test voltage. The steady-state inactivation of $I_{Ca}$ is reported as I/$I_{max}$, where I is the peak current in the final voltage step to –30 mV and $I_{max}$ is the maximum peak current for each cone. Data were sampled between 20 and 60 kHz and filtered at 3 kHz.

For horizontal cell light responses, horizontal slices were prepared from the central mouse retina. The identity of horizontal cells was determined based on their larger soma diameter (~15 µm) compared to bipolar cell somas (~6 µm). During whole cell patch clamp recordings, horizontal cells were held at –70 mV. Light stimuli (1 s) at 410 nm (630 × 830 µm) were presented onto the retina (at a minimum of 5 s intervals) through the microscope's condenser using a Polygon1000 DMD pattern illuminator (Mightex, Pleasanton, CA, USA) and a custom-built light path. Light intensity (in watts) was measured at the point on the microscope stage where the retina is placed using a power meter (Thorlabs, Newton, New Jersey, USA). Photon flux $\Phi_q$ (photons/s) within the light stimulus area was calculated using the measured light intensities in the formula:

$$\Phi_q \, \text{per} \, \mu m^2 = \frac{measured \, light \, intensity \, (W)}{\frac{hc}{\lambda}} / light \, stimulus \, area \, \left( \mu m^2 \right)$$

where $h$ is Planck's constant (J*s), $c$ is the speed of light (m/s), and $\lambda$ is the wavelength (m). Light stimulus intensity was increased in Log2 steps from $4.9 \times 10^2$ to $2.1 \times 10^5$ $\Phi_q$/µm². For each light intensity step, both ON and OFF current amplitudes were measured from baseline (averaged 5 ms of current prior to light onset) to the maximum positive (after light onset) or maximum negative (after light offset) current, respectively. To confirm the identity of these horizontal cell light responses as AMPA-mediated currents, 1 s light stimuli (410 nm, $1.2 \times 10^5$ $\Phi_q$/µm²) were continuously delivered every 10 s before, during, and after bath application of 20 µM DNQX.

For voltage-clamp recordings of 13-lined ground squirrel cones, retinal slices were prepared as previously described (*DeVries and Schwartz, 1999*). The eyecup was divided along the dorsal to ventral axis into superior, middle, and inferior parts. The dorsal area above the line of the optic nerve head was defined as superior, the central region with a width of about 5 mm just ventral to the optic nerve head was middle, and the remaining ventral area was inferior. Isradipine and ML218, alone or in combination, were applied from separate puffer pipettes whose orifices were aimed at the cone synaptic region. Recordings were made with an Axopatch 200B amplifier (Molecular Devices). Signals were electronically filtered at 5 kHz and digitized at a rate of 10 kHz. Additional Gaussian filtering was added (cutoff frequency of 500 Hz). Tissue was viewed through a 63 x water immersion objective on a Zeiss Axioskop FS2 microscope and superfused with an extracellular solution at room temperature. Drugs and their concentrations used during these experiments are described in *Table 3*. Membrane

potential was continuously maintained at –85 mV. During ramp stimulation, the membrane potential was depolarized to –85 to +35 mV at a rate of 1 mV/ms.

For rhesus macaque cone terminal recordings, a 12-year-old male was sedated with ketamine (5 mg/kg I.M.) and dexmedetomidine (0.015 mg/kg I.M.). Post-sedation, the animal received buprenorphine (0.02 mg/kg I.M.), atropine (0.02 mg/kg I.M.), and maropitant citrate (1 mg/kg S.Q.). The animal was intubated and maintained with inhaled isoflurane (0.75–2.0%) and propofol (7–8 mg/kg/hr I.V.). Crystalloid fluids (5 mL/kg/hr) were administered I.V., and phenylephrine (5–10 mcg/kg/hr I.V.) was used for blood pressure support. The animal was under anesthesia for approximately 3 hr prior to perfusion. Transcardial perfusion was approached through a midline thoracotomy. Immediately prior to perfusion, 5 mL of Euthasol (sodium pentobarbital 390 mg/mL/phenytoin sodium 50 mg/mL) was administered I.V. The descending thoracic aorta was clamped, the pericardium was opened, and the right atrium was cut. The apex of the left ventricle was sharply incised, and a large bore cannula (Yankauer suction handle, 5 mm internal diameter) was inserted through the left ventricle until it could be palpated in the ascending aorta. The cannula was clamped in place at the apex of the left ventricle. The animal was perfused with 4 L of cold phosphate-buffered saline at ~500 mL/min. Eyes were removed approximately 1 hr following perfusion. Eyes were dissected, and eye cups were allowed to dark adapt for ~30 min and stored in bicarbonate buffered Ames' media at 32 °C equilibrated with 95% $O_2$/5% $CO_2$ prior to slice preparations. Vertical sections of central retina were prepared from 5 mm retina punches as described for mice. Depending on the species, the electrophysiological experiments were performed one or more times with cells from one animal representing technical replicates.

## Single cell RNA sequencing

Retinas from 2 WT and 2 G369i KI mice (2–3 months old, all males) were dissected in ice-cold Dulbecco's Phosphate Buffered Saline (D-PBS) and dissociated into single cells using the Papain Dissociation System (Worthington Biochemical). Two retinas were incubated in a 2.5 ml of dissociation solution (40 U/ml papain, 2 mM L-cysteine, 200 U/ml DNAse in Earle's Balanced Salt Solution, EBSS) for 8 min at 37 °C. Following aspiration, 2 ml of inactivation solution (ovomucoid protease inhibitor and bovine serum albumin (BSA), 10 mg/ml each in EBSS) was added prior to trituration with a series of fire-polished, siliconized glass Pasteur pipettes. The solution was passed through a 40 μm mesh (pluriStrainer, Cat# 43-50040-51, pluriSelect) prior to centrifugation (200 × *g* for 5 min, 4 °C). Following aspiration of the supernatant, the cell pellet was resuspended in 0.5 ml D-PBS +0.5% BSA. Cell suspensions (10,000 cells per sample) were loaded on the Chromium Controller (10 X Genomics) and processed for cDNA library generation following the manufacturer's instructions for the Chromium NextGEM Single Cell 3' Reagent Kit v3.1 (10X Genomics). The resulting libraries were examined for size and quality using the Bioanalyzer High Sensitivity DNA Kit (Agilent) and their concentrations were measured using the KAPA SYBR Fast qPCR kit (Roche). Samples were sequenced on the NovaSeq 6000 instrument (paired end, read 1:28 cycles, read 2: 90 cycles) with a targeted depth of 9 K reads/cell.

scRNA-seq Fastq files were processed with CellRanger (v7.1.0) using the reference genome mm10–2020 A with introns included. CellRanger outputs were imported to Seurat (v5.0.1), a gene by feature count matrix was constructed, log normalization was performed, and the log normalized values were used for downstream analysis. Unsupervised clustering was performed in Seurat using resolution = 0.5. DEsingle method (v1.6.0, *Miao et al., 2018*) was used for differential expression analysis comparing G369i KI to WT in each cluster.

## Serial block-face scanning electron microscopy and 3D reconstructions

Eye cups were prepared from P42 WT and G369i KI littermates and fixed using 4% glutaraldehyde in 0.1 M cacodylate buffer, pH 7.4, for 4 hr at room temperature followed by additional fixation overnight at 4 °C. Glutaraldehyde-fixed eye cups were then washed 3 times in 0.1 M cacodylate buffer. Retinas were thereafter isolated and embedded in Durcupan resin after staining, dehydration, and embedding as described previously (*Della Santina et al., 2016*). A Thermo Scientific VolumeScope serial block face scanning electron micrscope was used to image embedded retinas. Retinal regions comprising a 2x2 montage of 40.96 μm tiles were imaged at a resolution of 5 nm/pixel and section thickness of 50 nm. Image stacks were aligned, and cone photoreceptor terminals were reconstructed

using TrakEM2 (NIH). Postsynaptic partners at cone ribbons were followed to the inner nuclear layer to determine their identity. Amira software was used for 3D visualization of reconstructed profiles.

## Electroretinography

Retinal function was assessed using the Celeris system (Diagnosys, Inc) paired with the Espion software (Diagnosys, Inc). Mice were anaesthetized under red light (660 nm) via intraperitoneal (I.P.) injection with ketamine/xylazine mixture (100 mg/kg ketamine, 10 mg/kg xylazine). Tropicamide ophthalmic solution (1 %) and hypromellose lubricant eye gel (0.3%) were administered topically to both eyes before the mouse was secured to a heated (37 °C) platform to maintain body temperature. Ag/AgCl corneal stimulators were placed on each eye. After collecting data from individual mice, atipamezole (Antisedan, 1–2 mg/kg) was I.P. injection administered to reverse the effects of the ketamine and xylazine. For photopic ERGs, eyes were light adapted using a background green light at 20 cd·s/m$^2$ for 10 min. Following light adaptation, eight different light intensity pulses (–0.5, 0, 0.5, 1, 1.5, 2, 2.3, and 2.6 log cd·s/m$^2$) were delivered on top of the background green light. ERG a-waves were measured from baseline to the peak of the negative potential. ERG b-waves were measured without filtering the oscillatory potentials from the peak of the a-wave to the peak of the positive potential within 60–70 ms (WT) and 80–100 ms (G369i KI and Ca$_V$1.4 KO) of the light stimuli. For flickering ERGs, mice were dark-adapted (>12 hr). White light pulses from –4–2.5 log cd·s/m$^2$ were delivered in 0.5 log unit steps at 10 Hz. Response amplitudes were measured from the trough to the peak of each response at all light intensities. Each intensity stimulus was delivered 10 times, with a 3 s interval between each stimulus, and averaged. The mean response amplitudes recorded in the right and left eye of each mouse are reported for all quantified ERG data. Experiments were performed at least three times with at least 3 mice per genotype (each mouse represented a biological replicate).

## Visible platform swim test

The visible platform swim test was performed as has been previously described (*Bimonte-Nelson et al., 2015*). WT, G369i KI, and Ca$_V$1.4 KO male and female mice (6–9 weeks old) were dark-adapted overnight and allowed to adapt to the procedure room for at least 1 hr prior to beginning the experiments. A water-filled, four-foot diameter galvanized steel tank and a visible white platform with a diameter of 10 cm was used for the swim test. Mice were subjected to six swim trials per day. Assays conducted under photopic (55 lux) and scotopic conditions (0 lux) were performed on days 1 and 2, respectively. Light intensity for photopic and scotopic conditions was measured at the platform using an Extech HD450 light meter (FLIR Systems, Nashua, New Hampshire). Mice were given 90 s to find the platform before being removed. After one trial was performed on all mice, the platform was moved to one of three different locations, top left, top center, and top right in relation to the initial site of mouse placement in the tank. Each trial was recorded using an infrared camera (Basler AG, Ahrensburg, Germany) and EthoVision XT16 software (Noldus Information Technology). Locating the platform was considered successful when mice contacted the platform with a head-on approach, even if mice failed to escape onto the platform. Mice were allowed to rest on the platform for 15 s at the end of each trial before being returned to a pre-warmed cage. Latency to find the platform was manually recorded and confirmed using recorded videos. Male and female mice were tested separately, and no sex differences in performance were identified post-hoc using two-way RM-ANOVA (*Supplementary file 1*). Male and female data were combined into a single group for each genotype. Latency to the platform was calculated by averaging the final 3 trials under scotopic and photopic conditions for each mouse and compared using Kruskal-Wallis one-way ANOVA with Dunn's posthoc multiple comparison test. Experiments were performed at least three times with at least 3 mice per genotype (each mouse represented a biological replicate).

## Ca$^{2+}$ imaging of cone terminals

Retinas were dissected and stored in an oxygenated mouse retina-slicing solution. The loading of the cell-permeant Ca$^{2+}$ dye Fluo-3, AM was modified from on a previous publication (*Johnson et al., 2007*). To label the base of cone terminals, retinas were incubated in retina slicing solution containing PNA Lectin conjugated with CF568 (100 $\mu$g/mL) for 15 min at room temperature. Then the retinas were incubated with retina slicing solution containing 1 mM Fluo-3, AM, and 0.025% pluronic acid (Biotium) for 45 min at room temperature followed by three rinses with retina slicing solution. The

retinas were then incubated in retina slicing solution for 15 min at 35 °C to increase esterase activity. Vertical retina slices were prepared as described above. Slices were mounted on a Hyperscope multi-photon system (Scientifica, Uckfield, United Kingdom) equipped with a resonant scanner and a Mai Tai HP DeepSee laser (Spectra-Physics, Milpitas, CA, USA) tuned to 800 nm at 5% power. Slices were perfused with normal extracellular solution containing (in mM) 112 NaCl, 3 KCl, 1 $MgCl_2$, 3 $CaCl_2$, 1.25 $NaH_2PO_4$, 10 Na Pyruvate, 22.5 $NaHCO_3$, 5 HEPES, 10 Glucose, osmolarity 300 ± 5 mOsm/kg. High-K + extracellular solution contained (in mM) 65 NaCl, 50 KCl, 1 $MgCl_2$, 3 $CaCl_2$, 1.25 $NaH_2PO_4$, 10 Na Pyruvate, 22.5 $NaHCO_3$, 5 HEPES, 10 Glucose, osmolarity 300 ± 5 mOsm/kg. Extracellular solutions were supplemented with 20 µM CNQX, 25 µM L-APV, and 100 µM picrotoxin. Isradipine (10 µM) was added to High-K + extracellular solution. Images (512 × 512) were collected as 15-frame averages at 0.93 Hz using ScanImage (MBF Bioscience, Williston, VT, USA). Changes of Fluo-3 fluorescence in cone terminals were measured in ROIs drawn juxtaposed to PNA-CF568 labeling using Fiji (*Schindelin et al., 2012*). Experiments were performed at least three times with at least 3 mice per genotype (each mouse represented a biological replicate).

## Data analysis

Electrophysiological data were analyzed by custom routines written in IgorPro software (Wavemetrics) and statistical analysis was performed using Prism software (GraphPad). Data were analyzed for normality by Shapiro Wilk test followed by parametric (t-test or ANOVA) or non-parametric methods (Kruskal Wallis, Mann-Whitney, Wilcoxon).

## Acknowledgements

J W M, G J O, J D V, A H, K R, and A L were supported by NIH grants EY026817 (to A L), EY029953 (to J W M), and unrestricted funds from the University of Texas-Austin (to A L); S R W and M H were supported by NIH grant EY031677 (to M H), an unrestricted grant from RPB to UW Madison Dept. of Ophthalmology and a McPherson ERI professorship to M H; D F and S H D were supported by NIH grants EY012141 and EY032506, an unrestricted grant from RPB to the Northwestern University Dept. of Ophthalmology, and an RPB International Travel Award. N A S and R D Mayfield were supported by AA029955 (N A S) and AA0012404 and AA020926 (R D M). The authors thank J Sun for assistance with acquiring some of the electrophysiological data in *Figure 3—figure supplement 1*, S Knecht and R Wong for assistance with serial EM image collection, E Seidemann and B Shukla for the generous donation and collection of macaque retinal tissue, respectively, M Agosto for mGluR6 and TRPM1 antibodies, and M McCall and F Vinberg for advice on ERGs.

## Additional information

### Funding

| Funder | Grant reference number | Author |
| --- | --- | --- |
| National Institutes of Health | EY026817 | Amy Lee |
| National Institutes of Health | EY029953 | J Wesley Maddox |
| National Institutes of Health | EY031677 | Mrinalini Hoon |
| National Institutes of Health | EY012141 | Steven DeVries |
| National Institutes of Health | EY032506 | Steven DeVries |
| National Institutes of Health | AA029955 | Nihal A Salem |
| National Institutes of Health | AA0012404 | Dayne Mayfield |

| Funder | Grant reference number | Author |
|---|---|---|
| National Institutes of Health | AA020926 | Dayne Mayfield |

The funders had no role in study design, data collection and interpretation, or the decision to submit the work for publication.

## Author contributions

J Wesley Maddox, Data curation, Formal analysis, Conceptualization, Investigation, Methodology; Gregory J Ordemann, Juan AM de la Rosa Vázquez, Boris V Zemelman, Formal analysis, Conceptualization, Methodology; Angie Huang, Kate Randall, Conceptualization; Christof Gault, Serena R Wisner, Daiki Futagi, Formal analysis, Conceptualization; Nihal A Salem, Writing – review and editing, Formal analysis; Dayne Mayfield, Supervision, Funding acquisition; Steven DeVries, Mrinalini Hoon, Formal analysis, Supervision, Funding acquisition, Conceptualization, Methodology; Amy Lee, Data curation, Resources, Formal analysis, Supervision, Funding acquisition, Writing - original draft, Project administration, Methodology

## Author ORCIDs

J Wesley Maddox ⓘ http://orcid.org/0000-0002-1630-2746
Boris V Zemelman ⓘ https://orcid.org/0000-0003-2463-4887
Amy Lee ⓘ https://orcid.org/0000-0001-8021-0443

## Ethics

All animal experiments were performed in accordance with guidelines approved by the National Institutes of Health and the Institutional Animal Care and Use Committees at the University of Texas at Austin and Northwestern University were approved by the Institutional Animal Care and Use Committee (AUP-2023-00170).

Public review (joint version of all reviewers) https://doi.org/10.7554/eLife.94908.4.sa1
Author response https://doi.org/10.7554/eLife.94908.4.sa2

---

# Additional files

## Supplementary files

- MDAR checklist
- Supplementary file 1. Statistical analysis of visible platform swim test disaggregated by sex.

## Data availability

All data generated or analysed during this study are included in the manuscript and supporting files; source data files have been provided for Figures 2–10, Figure 2—figure supplement 1, and Figure 3—figure supplement 1.

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
