## [Editor Report · eLife assessment]

Based on analyses of retinae from genetically modified mice, and from wild-type ground squirrel and macaque, employing microscopic imaging, electrophysiology, and pharmacological manipulations, this **valuable** study on the role of Cav1.4 calcium channels in cone photoreceptor cells (i) shows that the expression of a Cav1.4 variant lacking calcium conductivity supports the development of cone synapses beyond what is observed in the complete absence of Cav1.4, and (ii) indicates that the cone pathway can partially operate even without calcium flux through Cav1.4 channels, thus preserving behavioral responses under bright light. The evidence for the function of Cav1.4 protein in synapse development is **convincing** and in agreement with a closely related earlier study by the same authors on rod photoreceptors. The mechanism of compensation of Cav1.4 loss by Cav3 remains unclear but appears to involve post-transcriptional processes. As congenital Cav1.4 dysfunction can cause stationary night blindness, this work relates to a wide range of neuroscience topics, from synapse biology to neuro-ophthalmology.

---

## [Referee Report · Public review (joint version of all reviewers)]

Cav1.4 calcium channels control voltage-dependent calcium influx at photoreceptor synapses, and congenital loss of Cav1.4 function causes stationary night blindness CSNB2. Based on a broad portfolio of methodological approaches - genetic mouse models, immunolabeling and microscopic imaging, serial block-face-SEM, ERGs, and electrophysiology - the authors show that cone photoreceptor synapse development is strongly perturbed in the absence of Cav1.4 protein, and that expression of a nonconducting Cav1.4 channel mitigates these perturbations. Further data indicate that Cav3 channels are present, which, according to the authors, may compensate for the loss of Cav1.4 calcium currents and thus maintain cone synaptic transmission. These data, which are in agreement with a similar study by the same authors on rod photoreceptor synapses, help to explain what functional defects exactly cause CSNB2 and why it is accompanied by only mild visual impairment.

The strengths of the present study are its conceptual and experimental soundness, the broad spectrum of cutting-edge methodological approaches pursued, and the convincing differential analysis of mutant phenotypes. Weaknesses mainly concern the fact that the mechanism by which Cav3 channels might partially compensate for the loss of Cav1.4 calcium currents remains unclear.

---

## [Author Response]

The following is the authors’ response to the previous reviews.

Intro.47-48 rewrite sentence

This sentence has been rewritten as: Photoreceptor synapses are specialized with a vesicle-associated ribbon organelle and postsynaptic neurites of horizontal and bipolar cells that invaginate deep within the terminal

ResultsMajor comment. Lines 100-103The new rod data presented here looks like an n = 1. Neither the Results section nor Supp Fig S1, describe the number of cells used. Nor do the authors offer a statistical description with averages, etc.. In addition, the single traces are much improved over their previous study (Maddox et al eLife 2020), but the authors have not described any new approach or trick that improved their rod Ica. Neither Methods section nor Supp section describes the procedure for patching rods (solutions, or Vh which is critical for assessing T-type currents).Suggestion, if more data exists, then present it. Otherwise, drop the argument.

The recording methodology for recording rods was like that for cones and this has been clarified in the Methods section (lines 725-752). Averaged data (n = at least 5 per group) and statistical analyses have been added to Fig.S1 (renamed Figure 2-Figure Supplement 1), and clearly show that no Ca2+ currents are present in the KI rods.

Supp Fig S2. The legend needs to be fixed. Conversion to PDF file may have created these formatting errors.

This has been corrected (renamed Figure 3-Figure Supplement 2).

Fig 8 a. The position of the light stimulus bar in the KO panel appears to be out of place, shifted too far to the left.

This has been corrected.

Major comments. 219-221The use of Fluo3-AM is not properly stated here. The text reads "cone pedicles filled with the Ca2+ indicator Fluo3". The wording used could be wrongly interpreted as: whole-cell filling of the cones via patch electrode. However, the Methods section describes bathing the retina in Fluo3-AM, which presumably fills PRs, HCs tips, Mueller glia and bpc dendrites. The Results section should acknowledge that the retina was loaded with Fluo3-AM.The cell types, and their processes (Muellers, HCs, bpc, PRs), present in a cone pedicle ROI will likely contribute to the Fluo3 readout of Ca2+ in the OPL, because (1) the EM images in Fig 7 highlight how interdigitated the processes are with the presynapse, (2) all express Cav channels, and many if not all express L-Type Cavs in their processes (glia, HC, on-bcs and PRs), and (3) all are depolarized with the addition of high extracellular KCl. The inclusion of Isradipine will inhibit L-type Cavs on pre- and post-synaptic targets, failing to specifically isolate PR Ca2+. Furthermore, Glu Receptor blockers are used here, which would be a great idea if the cones were stimulated with light; however, KCl bypasses the excitatory synaptic pathway and depolarizes all processes within the ROI. Hence, all cellular parts in the ROI will potentially contribute to Fluo3-Ca2+ signals.Suggestions for presentation of these findings. Ultimately your conclusion is suitable " 233 to 234...... Taken together, our results suggest that Cav3 channels nominally support Ca2+ signals and synaptic transmission in cones of G369i KI mice". The dramatic reduction in Fluo3-Ca2+ signals in the OPL G369i retinas (Fig 9) is a valuable finding for the following reasons: (1) the results do not show a clear compensation from intracellular stores that could potentially supersede the T-type currents in the G369i (which is an argument you make), and (2) there is a massive loss of Ca2+ influx in the OPL of G369i retinas. Since G369i is specific to the PRs, and only cones are present in the mutant G369i, the loss of Fluo3-Ca2+ signal in the mutant ROI reflects in large part loss of cone Fluo3-Ca2+ signals. Your findings illustrate the severity of the mutation, which has also been addressed in the various electro-physio sections of the MS.Figure 9 also needs to be more clear about (1) the loading of the cells with AM-dye, and (2) the presence of glia, HCs and bc dendrites in the PNA demarcated ROIs.

We regret that we did not make this more clear, but our Fluo 3 loading protocol of whole retina followed by vertical slicing allowed for loading primarily of photoreceptors in the portion of the outer retina that we imaged. We clarified this with the following edit to the text (lines 220-226):

“To test if the diminished HC light responses correlated with lower presynaptic Ca2+ signals in G369i KI cones, we performed 2-photon imaging of vertical slices prepared from whole retina that was incubated with the Ca2+ indicator Fluo3-AM and Alexa-568-conjugated peanut agglutinin (PNA) to demarcate regions of interest (ROIs) corresponding to cone pedicles. With this approach Fluo3 fluorescence was detected only in photoreceptors and ganglion cells and not inner retinal cell-types (e.g., horizontal cells, bipolar cells, Mueller cell soma). Thus, Ca2+ signals reported by Fluo3 fluorescence near PNA-labeling originated primarily from cones.”

We also note that given the considerably larger volume of the cone pedicle relative to the postsynaptic neurites of horizontal and bipolar cells, as well as neighboring glia, it seems unlikely that the latter would contribute significantly to the isradipine-sensitive Ca2+ signal measured in the ROI above the PNA labeling. Moreover, to our knowledge the contribution of Cav1 L-type channels to postsynaptic Ca2+ signals in the dendritic tips of horizontal cells and bipolar cells has not been demonstrated.